# Autocatalytic and oscillatory reaction networks that form guanidines and products of their cyclization

Alexander I. Novichkov[1], Anton I. Hanopolskyi[1], Xiaoming Miao[1], Linda J. W. Shimon [2], Yael Diskin-Posner [2] & Sergey N. Semenov [1✉]

Autocatalytic and oscillatory networks of organic reactions are important for designing life-inspired materials and for better understanding the emergence of life on Earth; however, the diversity of the chemistries of these reactions is limited. In this work, we present the thiol-assisted formation of guanidines, which has a mechanism analogous to that of native chemical ligation. Using this reaction, we designed autocatalytic and oscillatory reaction networks that form substituted guanidines from thiouronium salts. The thiouronium salt-based oscillator show good stability of oscillations within a broad range of experimental conditions. By using nitrile-containing starting materials, we constructed an oscillator where the concentration of a bicyclic derivative of dihydropyrimidine oscillates. Moreover, the mixed thioester and thiouronium salt-based oscillator show unique responsiveness to chemical cues. The reactions developed in this work expand our toolbox for designing out-of-equilibrium chemical systems and link autocatalytic and oscillatory chemistry to the synthesis of guanidinium derivatives and the products of their transformations including analogs of nucleobases.

[1] Department of Molecular Chemistry and Materials Science, Weizmann Institute of Science, Rehovot, Israel. [2] Department of Chemical Research Support, Weizmann Institute of Science, Rehovot, Israel. ✉email: sergey.semenov@weizmann.ac.il

Chemical clocks are centrally important to cellular life[1]. Regular oscillations in the concentrations of ions and biomolecules regulate cellular metabolism, signaling, and division, and, on a larger scale, control circadian rhythm, heartbeat, and intestinal contractions[2–8]. These clock-like fluctuations result from dynamic changes in transcription and translation or—perhaps, less commonly—enzymatic activity (e.g., glycolytic fluctuations and circadian rhythm)[4,5,9].

If introduced into artificial cells[10–15] and life-inspired materials[16–29], chemical oscillators function as autonomous time-keeping systems. Interaction with periodically changing environment (e.g., day and night cycles), periodic movements (e.g., contractions and expansions), restructuring, or release of chemicals all require internal clocks[26].

Intriguingly, despite their ubiquitous biological function, chemical oscillators remain difficult to recreate in synthetic systems. Inorganic oxo-halogen-based oscillators form the biggest class of chemical oscillators[30–33], but they involve strongly oxidizing and/ or extreme pH conditions and lack the structural versatility of either biochemistry or synthetic organic chemistry. The groups of Huck[34,35], Rondelez[36,37], and Winfree[38] designed several synthetic biochemical oscillators. Surprisingly, synthetic organic molecules—the molecules over whose structures we have the best control—are the most challenging substrates for the development of oscillatory systems. Thioesters-based oscillators remain the only example of homogenous oscillators where all reactants and products are organic molecules[39,40].

Nevertheless, oscillators based on organic molecules offer various advantages. The structural versatility of organic molecules aids in controlling the oscillations (e.g., period and amplitude) by tuning the rate constants through the structures of reactants[29,40–44]. Organic molecules can be fluorescent, colored, and biologically active. They can form polymers with a variety of mechanical properties and biocompatibilities. Moreover, organic molecules interface naturally with the dynamic supramolecular polymers recently developed by van Esch, Eelkema, and Boekhoven[20–23]. Finally, organic oscillators might teach us about possible roles of chemical rhythms in prebiotic evolution.

In this work, we diversified the chemistry of organic oscillators. First, we developed thiol-assisted formation of guanidines from S-alkyl isothiouronium salts (we will call them thiouronium salts for simplicity in the rest of the manuscript). Second, we obtained the autocatalytic version of this reaction. Third, we combined this autocatalysis with negative feedback to obtain an oscillatory reaction that forms guanidines periodically. Having guanidines as an output of the oscillator opens possibilities to couple the oscillator's output to the formation of polyelectrolyte assemblies (e.g., polyguanidines with polyphosphates) and to the downstream formation of biologically active heterocycles (e.g., folic acid, caffeine, and tetrodotoxin) and heterocycles involved in specific molecule recognition (e.g., guanine, cytosine, uracil, and their analogs) all of which can be formed by cyclization and hydrolysis of guanidines[45,46].

## Results

**Designing the thiol-assisted synthesis of guanidines.** We used an analogy between thiouronium salts and thioesters to design thiol-assisted synthesis of guanidines. Thiouronium salts are carbonyl derivatives that undergo $S_N2$ substitution reactions with the release of thiols as thioesters do. Thus, we hypothesized that substitution of one thiol by another in thiouronium salts (i.e., the thiol-thiouronium salt exchange reaction) would proceed orders of magnitude faster than the amination of thiouronium salts in water at neutral pH. When the reacting thiol is in the β or γ position to amine, the thiol-thiouronium salt exchange will be

followed by a fast S - > N amidine transfer reaction resulting in thermodynamically stable guanidinium, as Doherty demonstrated previously[47]. The overall sequence of the thiol-exchange followed by S - > N transfer reassembles the famous native chemical ligation (Fig. 1)[48].

The studied reactions between thiouronium salts and thiolamines are summarized in Fig. 2a, b. First, we tested our hypothesis by comparing the rates of the reaction of cysteamine (**1**) (50 mM) and ethanolamine (**2**) (80 mM) with the thiouronium salt derived from MESNA (2-mercaptoethane-1-sulfonic acid sodium salt) thiol (**3**) (50 mM) in phosphate buffer (PB) (Fig. 2c). Compounds **1** and **2** have comparable pKas (8.3 and 9.5) and both are expected to form guanidinium derivatives. $^1H$ NMR kinetics showed that the reaction of **1** with **3** was completed in about half an hour, whereas the reaction of **2** with **3** did not show any progress within 2 h. The product of the reaction between **3** and **1** is 2-mercaptoethylguanidine (**11**), as confirmed by NMR chemical shifts of methylene groups (at δ 3.32 ppm and 2.67 ppm)[49].

This experiment confirmed the hypothesis that the thiol group in the β position to amine will strongly (here more than 100-fold) accelerate the formation of guanidines from thiouronium salts in water.

In the next set of experiments, we investigated how the structure of thiouronium salts influenced the rate of the amination reaction (Fig. 2d). We tested three variations in the structure of thiouronium salt: (i) changing MESNA thiol to thiophenol (compound **4**) and ethanthiol (compound **5**); (ii) using a derivative of thiosemicarbazide (compound **6**); (iii) substitution of the amine group for the piperidinium group (compound **7**). Thiophenol is a better leaving group than MESNA thiol; thus, compound **4** reacted with **1** about 1.5 times faster than **3**. Similarly, ethanthiol is a worse leaving group than MESNA thiol, and compound **5** reacted slower than **3**. Thiosemicarbazide derivative **6** was more reactive than **5**. Introduction of a piperidinium group in **7** did not significantly influence the rate of guanidination compared to **5**; however, the product **13** undergoes further transformation. A similar transformation was observed for **11** but it required several days.

We investigated this follow-up transformation by studying the reaction between **4** and cysteine because its initial product, 1-

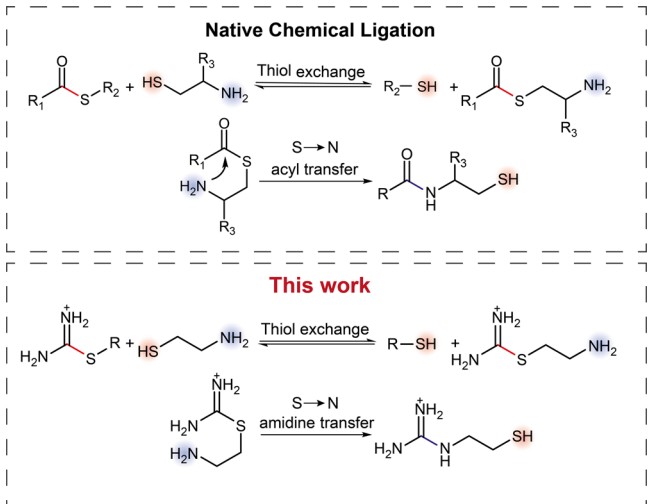

**Fig. 1 The analogy between native chemical ligation and the thiol-assisted formation of guanidines from thiouronium salts.** The top panel shows the mechanism of native chemical ligation. The bottom panel shows a proposed mechanism for the thiol-assisted formation of guanidines.

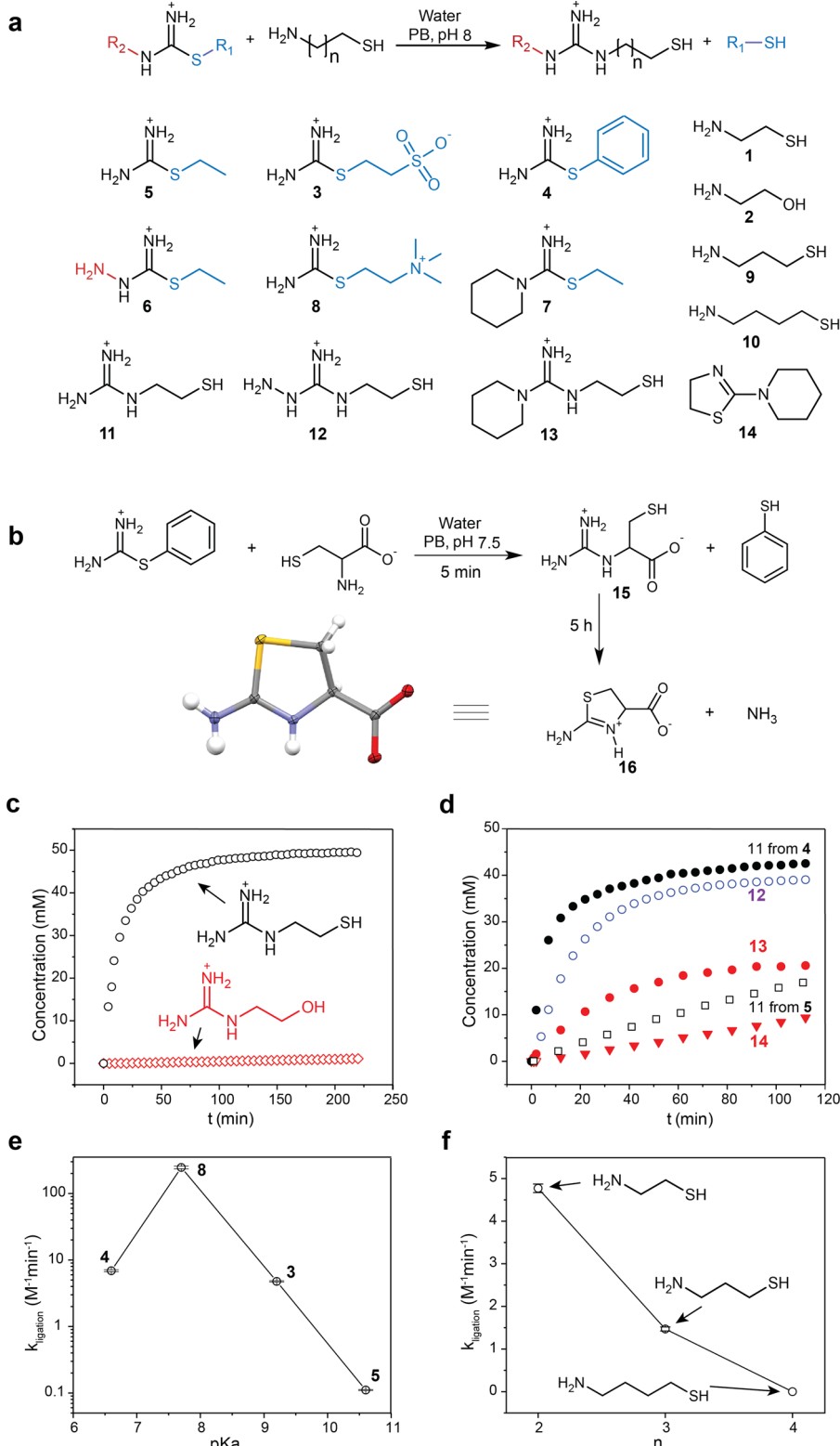

carboxy-2-mercaptoethylguanidine (**15**), rearranges faster than **11** (Fig. 2b). The single-crystal X-ray analysis revealed that the product of this rearrangement is carboxythiozolidine (**16**)[47,51]. Therefore, we concluded that **11** and **13** rearrange into thiazolidin-2-imine and compound **14** correspondingly.

To facilitate the further design of thiouronium salts, we analyzed the dependence of the rate constants of the reactions between thiouronium salts and **1** from the pKa of the leaving

group (Fig. 2e). The series included compounds **3–5** as well as a thiocholine derivative **8**[52]. The graph shows the expected negative trend between the pKa of a leaving group and the rate of nucleophilic substitution; however, thiophenol derivative **4** reacted slower than expected from the pKa of thiophenol (6.6)[53]. Most likely, this deviation is related to thiophenol being aromatic thiol in contrast to ethanethiol, MESNA thiol, and thiocholine.

**Fig. 2 Reactions of thiouronium salts with thiolamines. a** A general scheme of the reactions between thiouronium salts and thiolamines and the molecular structures of reactants and products studied in this work. **b** The reaction of **4** with cysteine and the subsequent cyclization of the guanidinium product to the thiazolidine product whose X-ray structure is shown. Ellipsoids are drawn at 50% probability. **c** [1]H NMR kinetics of the reactions of **3** with **1** and **2**. Reactions conditions: $D_2O$, 1 M PB pH 8.0, at 25 °C, [**3**] = 50 mM; [**1**] = 50 mM; [**2**] = 80 mM. **d** The effect of substituents of thiouronium salts on the rates of their reaction with **1**. Reactions were monitored by [1]H NMR. Reactions conditions: $D_2O$, 1 M PB pH 8.0, at 25 °C, ([**4**] = 50 mM; [**1**] = 54 mM), ([**5**] = 50 mM; [**1**] = 50 mM), ([**6**] = 42 mM; [**1**] = 50 mM), ([**7**] = 50 mM; [**1**] = 44 mM). **e** The effect of the pKa of the thiol leaving groups in thiouronium salts **3**, **4**, **5**, and **8** on the rate constants of their reactions with **1**. The rate constants were obtained by fitting NMR kinetic data with a numerical model using the COPASI program (Supplementary Section 4)[50]. Reactions conditions: $D_2O$, 1 M PB pH 8.0, 25 °C, ([**3**] = 50 mM; [**1**] = 50 mM), ([**4**] = 50 mM; [**1**] = 54 mM), ([**5**] = 50 mM; [**1**] = 46 mM), ([**8**] = 1 mM; [**1**] = 1 mM). With **8**, we lowered the concentrations of the reagents to reduce the reaction rate to a level measurable by [1]H NMR. **f** The effect of the length of the methylene bridge in thiolamines on the rate constants of their reactions with **3**. The rate constants were obtained following the same procedure as in (**e**). Reactions conditions: $D_2O$, 1 M PB pH 8.0, 25 °C, [**3**] = 50 mM; [**1**] = 50 mM; [**9**] = 50 mM; [**10**] = 50 mM. Error bars in **f** and **e** represent standard deviations.

To investigate the selectivity of this reaction to β-thiolamines, we varied a number of methylene groups between the amine and thiol groups (Fig. 2f). We synthesized γ-aminothiol (**9**) and δ-aminothiol (**10**) and reacted them with **3**. Compound **9** reacted ~3 times slower than **1** because in the reaction of **9** with **3** the intramolecular rearrangement undergoes a six-member cyclic transition state that is entropically less favorable than the five-member cyclic transition state involved in the reaction of **1**. Importantly, we did not observe any follow-up cyclization to thiazinan imine in this reaction. Compound **10** did not exhibit any acceleration of the formation of guanidine, even when compared with **2**. This result indicates that the intramolecular rearrangement through the seven-member ring transition state does not contribute significantly to the formation of a guanidine product, probably because of the high enthalpy penalty in the strained seven-member cyclic transition state.

**Autocatalytic reactions of thiouronium salts with cystamine.** The strong acceleration of the formation of guanidines from thiouronium salts by the thiol group in the β or γ position to amines suggests that the reaction of thiouronium salts with the disulfides of β- or γ-aminothiols should proceed autocatalytically through a nucleophilic chain reaction (Fig. 3a)[39,40]. To test this hypothesis, we reacted thiouronium salts **3**, **4**, and **8** with cystamine (**17**) in PB pH 8. The autocatalysis is initiated by thiols released in hydrolysis and aminolysis of thiouronium salts. These thiols exchange with **17**, forming **1**. This initiation corresponds to the lag phase on kinetic curves (Fig. 3b). Next, a thiouronium salt reacts with **1**, producing two thiols: **11** and a thiol released from the thiouronium salt (i.e., thiophenol, MESNA thiol, or thiocholine). These two thiols exchange with **17** and produce a corresponding asymmetric disulfide and two equivalents of **1** (Fig. 3a). Thus, one molecule of **1** produces two molecules of **1**. This phase of the reaction corresponds to the exponential growth. At the end of the process, thiouronium salts and **11** are depleted and the reaction slows down. To confirm that the sigmoidal shape of the kinetic curves shown in Fig. 3b is caused by autocatalytic amplification, we performed standard addition experiments (Fig. 3c). We added 0.2 and 2 mol% of mercaptoethanol to a reaction of **4** (50 mM) with **17** (50 mM) in PB pH 8. The addition of 0.2% of mercaptoethanol shortened the lag phase, whereas 2% of mercaptoethanol eliminated it completely, confirming the autocatalytic nature of the reaction.

**Cyclization of the guanidines formed in the autocatalytic reaction.** One of the major chemical differences between amides and derivatives of guanidines is that the former relatively easy participate in various cyclization reactions that give heterocycles including analogs of nucleobases. To demonstrate cascade cyclizations of the autocatalytically formed guanidines, we synthesized disulfides **18** and

**19** by reacting correspondingly **17** and the disulfide of 3-aminopropane-1-thiol with acrylonitrile. Molecules **18** and **19** undergo autocatalytic reaction with thiouronium salt **4** (Fig. 4a, b). The autocatalytic profile of the reactions of **18** with **4** is similar to the profile of the reaction of **17** and **4** indicating that transition from primary to secondary amine has little influence on the ligation rate (Fig. 4c). The guanidines **20** and **21** further undergo cascade cyclization leading to bicycles **22** and **23** that contain dihydropyrimidine ring. Expectedly, the formation of **22** is much faster than the formation of **23**. The identity of the compounds **22**–**23** in the reaction mixture was confirmed by high-performance liquid chromatography—mass spectrometry (HPLC–MS) analysis and, for **22**, by comparing signals in [1]H NMR of the reaction mixture with signals of the cyclization product of 3-(2-iminothiazolidin-3-yl)propanenitrile (Supplementary Section 5). Evaporation of the reaction mixture resulted in the partial hydrolysis of **22** with formation of **24**, which was isolated by chromatography. The identity of **24** was confirmed by NMR and X-ray crystallography (Fig. 4b). Heating the slurry left after evaporation of the reaction mixture with $I_2$ provided the pyrimidine derivative **25** which shows two characteristic doublets in [1]H NMR at 7.81 and 5.83 ppm.

Iminothiazolidine, which forms from **11** upon heating, was another interesting target for the one-pot formation of heterocycles[54,55]. Thus, heating the products of the autocatalytic reaction between **4** and **17** with the subsequent addition of pyridine and malononitrile results in the formation of the aminopyrimidine derivative **26**, which was identified by comparison with a separately synthesized standard (Fig. 4d, f and Supplementary Section 5).

Guanidines themselves can cyclize into analogs of nucleobases, albeit it requires strongly basic conditions and purified materials. We reacted disulfide **27**, which is a part of the dynamic library of disulfides formed in the reactions of **17** with thiouronium salts, with ethyl acetoacetate (Fig. 4e). The reaction produced a mixture of isomers from which we chromatographically isolated the major product. [1]H-, [13]C-NMR, HRMS, and X-ray structural analysis revealed that this product is a derivative of 2-amino-6-methylpyrimidin-4(3H)-one (**28**), which is a synthon for hydrogen bonding. In a crystalline form, **28** adopts a cyclic conformation that is held by a double hydrogen bond between two pyrimidine fragments (N(5) ⋯ O(1) 2.629 Å, N(4) ⋯ N(3) 3.364 Å) (Fig. 4g). To form this hydrogen bond, the pyrimidine fragments in **28** adopt two different tautomeric forms: one that reassembles donor–donor–acceptor motif of guanine and another that reassembles acceptor–acceptor–donor motif of cytosine.

**From autocatalysis to oscillations.** Next, we designed oscillatory systems using the thiouronium salt-based autocatalysis. We took the structure of the oscillatory reaction network from the thioester-based oscillator[40]. This network consists of positive feedback based on autocatalysis coupled with two negative

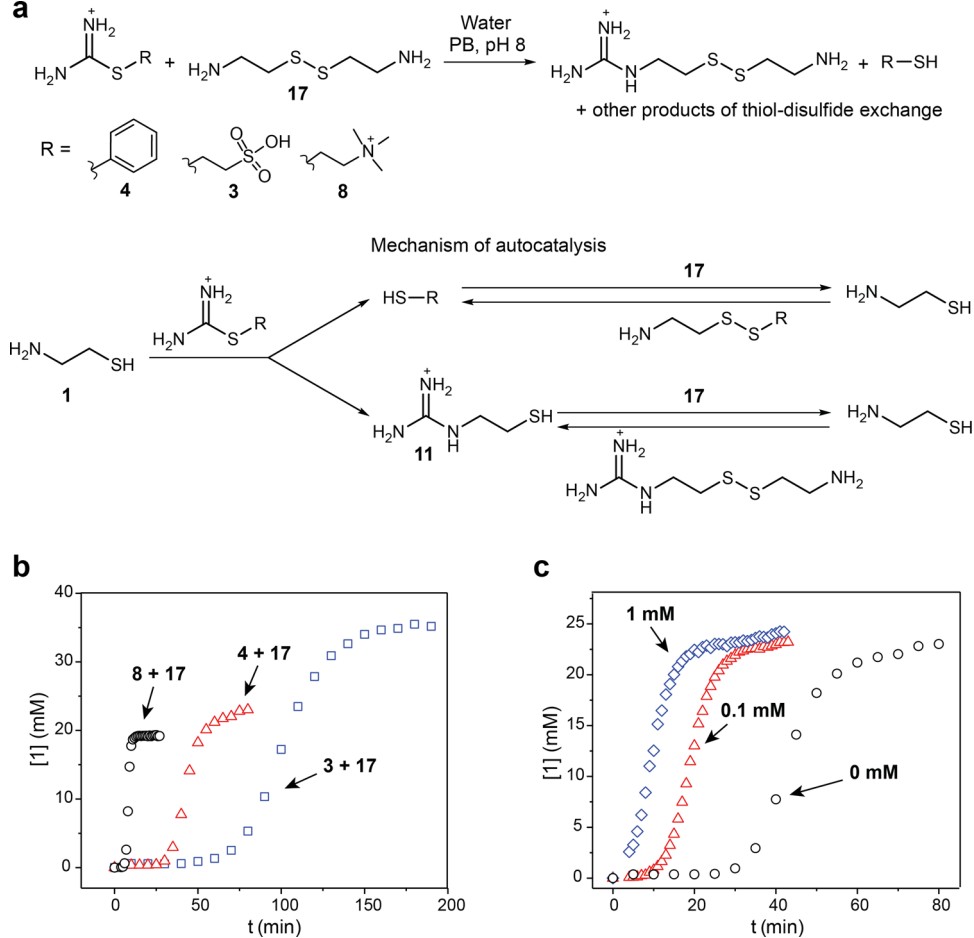

**Fig. 3 Autocatalytic reactions of thiouronium salts with cystamine (17). a** The general scheme of the reactions between thiouronium salts **3**, **4**, **8**, and **17**. The mechanism underlying the exponential growth of a concentration of **1**. **b** The kinetics of the reactions of **3**, **4**, and **8** with **17**. We monitored the kinetics of these reactions by following the production of **1** by $^1$H NMR spectroscopy. Reactions conditions: D$_2$O, 1 M PB pH 8, 25 °C, ([**3**] = 50 mM; [**17**] = 50 mM), ([**4**] = 50 mM; [**17**] = 50 mM), ([**8**] = 50 mM; [**17**] = 50 mM). **c** Experiments showing the elimination of the lag period in the autocatalytic reaction of **4** and **17** by addition of β-mercaptoethanol (0.1 and 1 mM). Reaction conditions: D$_2$O, 1 M PB pH 8, at 25 °C, [**4**] = 50 mM; [**17**] = 50 mM.

feedbacks (Fig. 5a). In the current design, we used a thiouronium salt-based autocatalysis as positive feedback, the reactions of thiols with maleimide or K$_3$[Fe(CN)$_6$] as the first negative feedback, and the reaction of thiols with acrylamide as the second negative feedback. Maleimide and acrylamide were used in the thioester-based oscillator; thus, they were the first choice. Both molecules react with thiols through conjugate addition, but the reaction with maleimide is much faster than with acrylamide (Fig. 5b). K$_3$[Fe(CN)$_6$], which almost instantly oxidizes thiols to disulfides, was used to demonstrate the diversity of chemistries applicable in this oscillatory network. A dramatic change in a chemical nature of the reaction—from nucleophilic addition to oxidation—does not affect the role of this reaction in the oscillatory network.

The design approach presented here also reassembles the strategy that was theoretically outlined by Boissonade and De Kepper and was experimentally used by Epstein and colleagues[32,56,57]. The method states that a bistable chemical system can be converted into oscillatory by the addition of an appropriate negative feedback species. For the oscillator in this manuscript, the negative feedback species is acrylamide. The bistable system is thiouronium salt-based autocatalytic network combined with maleimide or K$_3$[Fe(CN)$_6$].

Oscillators with topology used here require an open flow system to produce sustained oscillations. A continuously stirred tank reactor (CSTR) is the most controllable and easiest to numerically model flow reactor. To optimize the concentrations of the reactants, pH, and flow rates in CSTR, we used a numerical model previously developed for a thioester-based oscillator[40]. The modeling showed that out of all synthesized thiouronium salts, only **8** reacts sufficiently fast with **17** to produce oscillations (Supplementary Section 7). Next, we experimentally optimized conditions to produce a sharp pulse in batch (Supplementary Section 4). These experiments showed that the optimal conditions were **8** (50 mM), **17** (100 mM), maleimide (8 mM), acrylamide (300 mM), at pH 8 in 1 M PB.

Finally, the model predicted that the optimum ratio ($f/V$) of the flow ($f$) in the reactor to its volume ($V$) should be in the range of $1.2$–$2 \times 10^{-4}$ s$^{-1}$. The value of $f/V$, which is also called the space velocity, is a key parameter determining the kinetics of reactions in CSTR because it is an inverse of the average residence time of species in CSTR. We used a micro-CSTR set-up to conduct the flow experiments (Supplementary Section 6)[35,40]. The reagents were added to the CSTR using three syringes (Fig. 5c). The first syringe contained **17** in PB; the second syringe contained **8** in water; the third syringe contained acrylamide mixed with maleimide or K$_3$[Fe(CN)$_6$] in water. The CSTR was connected to the static microfluidic mixer where the content of CSTR was mixed with Ellman's reagent. The products of derivatization with Ellman's reagent were analyzed in an UV–vis flow cell or by HPLC–MS (Fig. 5c).

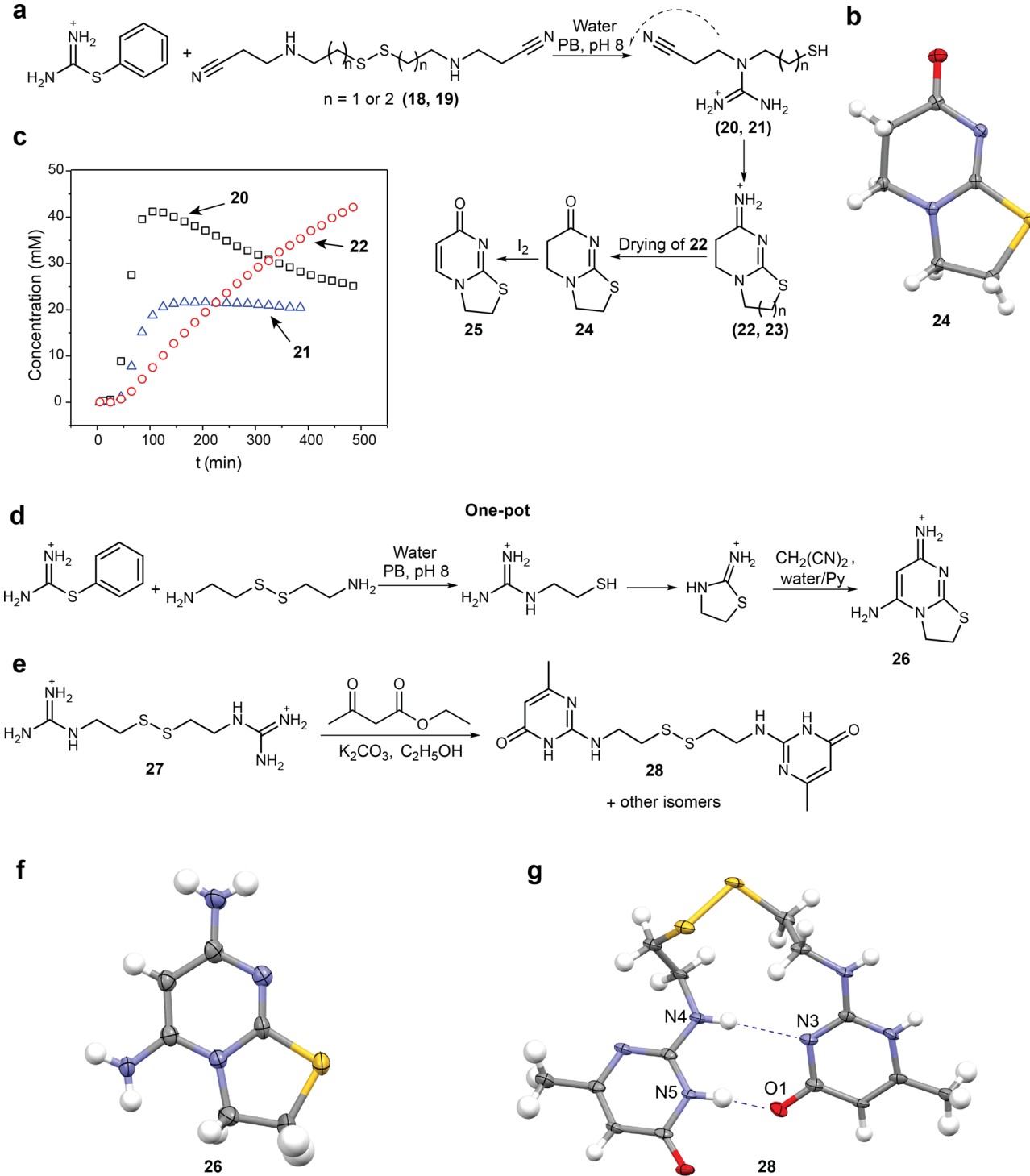

**Fig. 4 Cyclization of guanidines formed in autocatalytic reactions. a** Reactions of disulfides **18** and **19** with **4**. Compounds **20–23** form spontaneously in water at pH 8. Compounds **24–25** form in the one-pot process. **b** An ORTEP diagram for X-ray structure of **24** (50% probability ellipsoids). **c** The kinetics of the reactions of **18** and **19** with **4**. Concentrations of **20** and **22** were monitored in the reaction of **18** by $^1$H NMR. The concentration of **21** was monitored in a separate reaction with **19**. Reactions conditions: D$_2$O, 1 M PB pH 8, 25 °C, [**4**] = 70 mM; [**18 or 19**] = 70 mM). **d** One-pot formation of the aminopyrimidine **26** from **4**, **17**, and malononitrile. **e** Scheme of the reaction between disulfide **18** and ethyl acetoacetate. **f** An ORTEP diagram for X-ray structure of **26** (50% probability ellipsoids). **g** An ORTEP diagram for X-ray structure of **28** (50% probability ellipsoids). Hydrogen bonds are denoted as blue dashed lines.

We successfully observed sustained oscillations in experiments with both maleimide and K$_3$[Fe(CN)$_6$] (Fig. 6a, b). Oscillations were characterized by sharp peaks and flat regions separating them which is typical for substrate-depletion oscillators[39,40,58].

The best way to understand the mechanism underlying these oscillators is to individually discuss the different stages of the oscillatory cycle. At the first stage, when CSTR has been filled, the dominant reactions are the hydrolysis of **8**, which produces

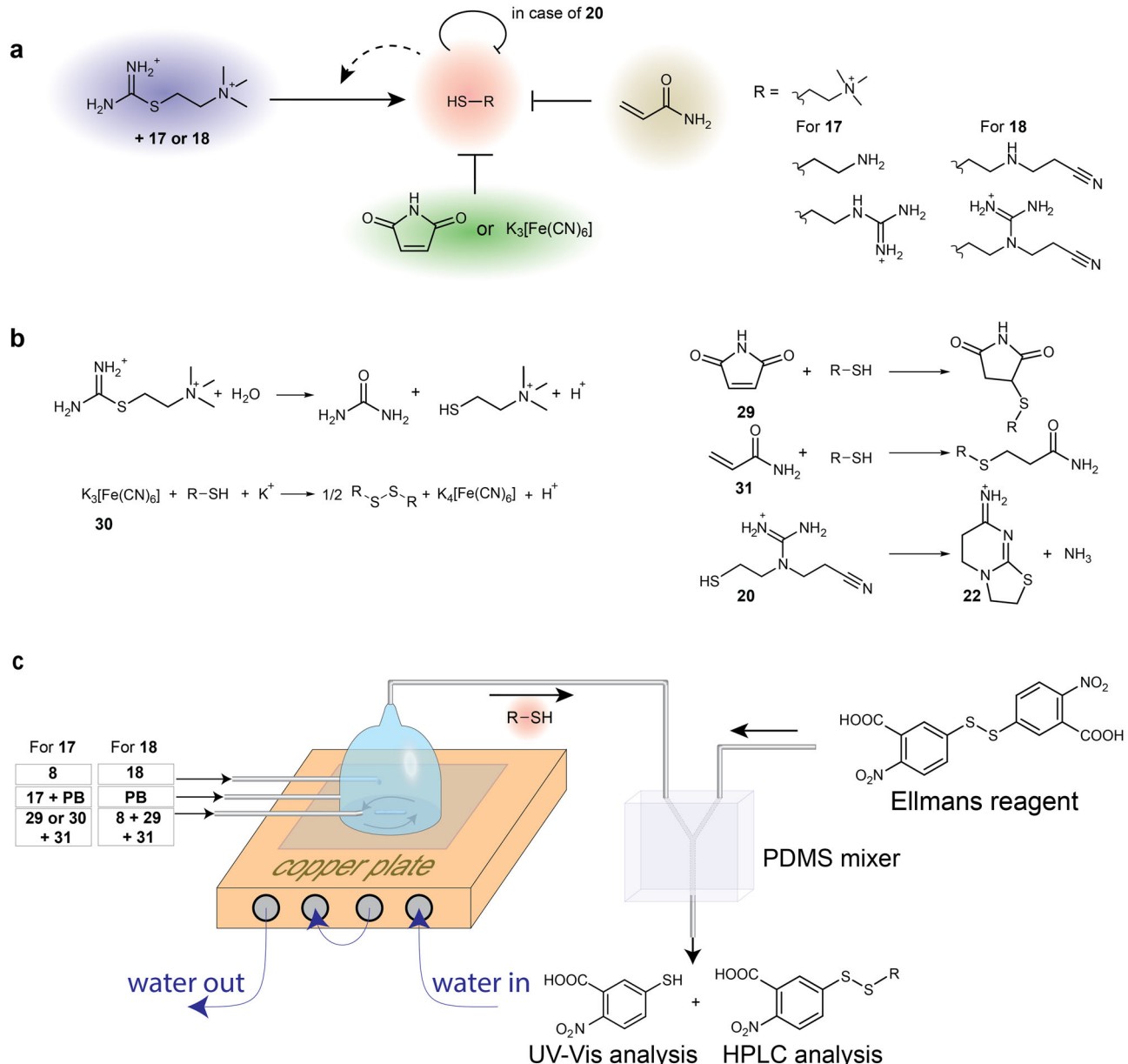

**Fig. 5 Thiouronium salt-based oscillators. a** A schematic layout of the oscillators based on the autocatalytic production of thiols from thiouronium salts and the suppression of this autocatalysis by maleimide or K$_3$[Fe(CN)$_6$] and acrylamide (negative feedbacks). In case of the oscillator with disulfide **18**, the autocatalysis is additionally suppressed by intramolecular cyclization **20**. **b** The schemes of the hydrolysis of **8** and, reactions of thiols with maleimide, K$_3$[Fe(CN)$_6$], and acrylamide, and cyclization of **20**. **c** A schematic representation of the CSTR experimental set-up and downstream derivatization with Ellman's reagent. The UV--vis spectrometer detects the absorption of 4-nitro-3-carboxythiophenolate at 412 nm. The HPLC separates and quantifies the mixed disulfides of 4-nitro-3-carboxythiophenole with **1**, **11**, and thiocholine. For analysis of oscillations in the concentration of **22**, the drops were directly collected from the outlet of the CSTR and analyzed by HPLC-MS.

thiocholine, and the reaction of thiocholine with maleimide or K$_3$[Fe(CN)$_6$]. The combination of the production and removal of thiocholine forms a trigger that initiates the autocatalytic production of **1** after some delay. At the second stage, when autocatalysis is initiated, the dominant processes are the autocatalytic production of thiols, the depletion of **8**, and the conjugate addition of thiols to acrylamide. When **8** is depleted, the autocatalysis halts and the trigger cannot set it again because of the low rate of the production of thiocholine from **8**. The third stage is dominated by the refilling of **8** and maleimide or K$_3$[Fe(CN)$_6$]. When the reagents are refilled, the next cycle starts and generates the second oscillation. Therefore, the

substrate depletion and the trigger are the main sources of instability in this oscillator in contrast to the majority of biological oscillators where delayed negative feedback is the source of instability[59].

The UV–vis flow cell provides information about the total concentration of all thiols in the system. To estimate the fraction of **1**, **11**, and thiocholine in the mixture of thiols, we collected drops of the solution coming out of the flow cell and analyzed them by HPLC, which clearly resolved assymetrical disulfides formed in reactions of **1**, **11**, and thiocholine with Ellman's reagent (Fig. 6c). The data indicate the prevalence of **1** and about equal amounts of **11** and thiocholine. This distribution is

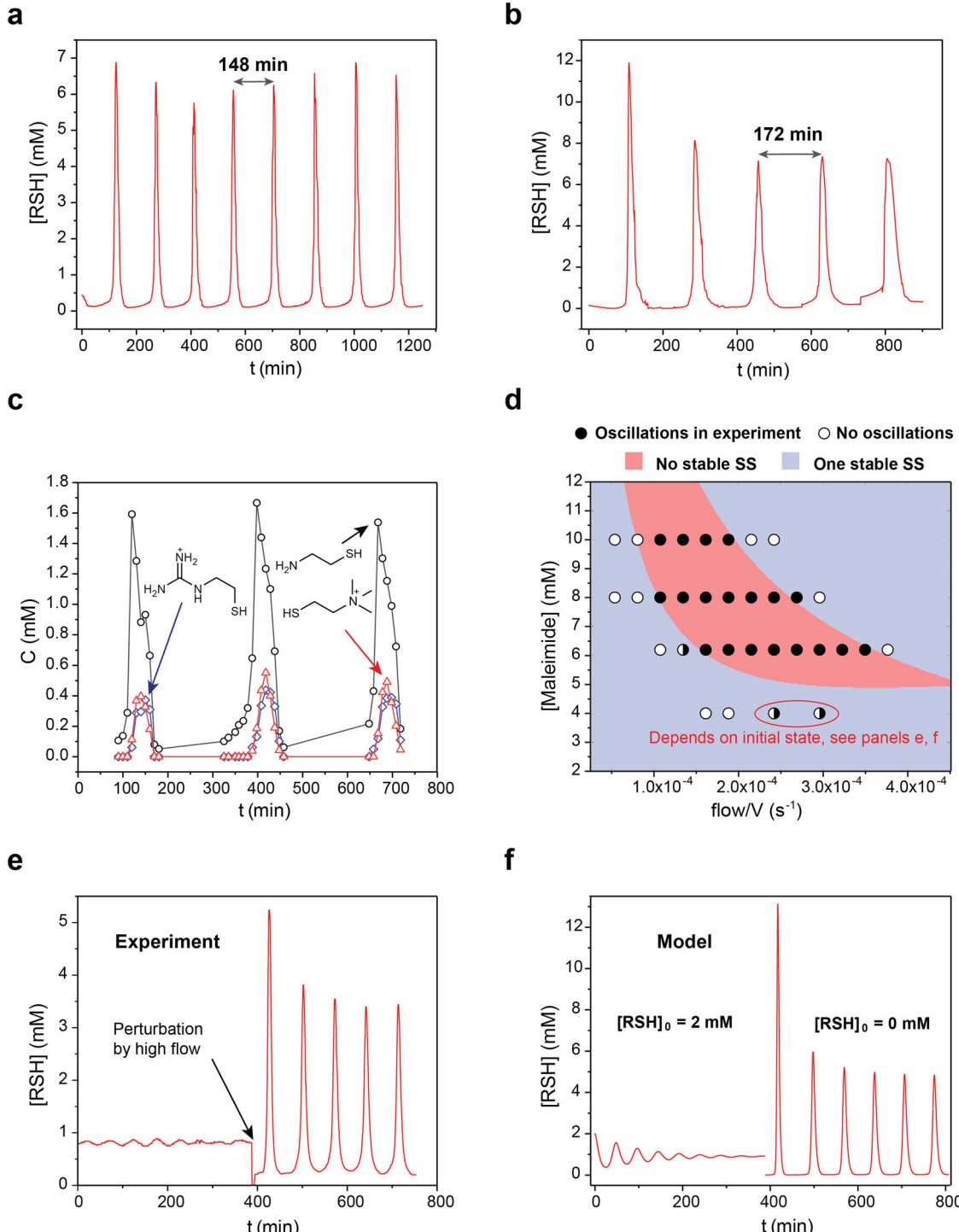

**Fig. 6 Kinetic studies of thiouronium salt-based oscillators.** Experimental conditions shared by all experiments: $H_2O$, 1 M PB pH 8, at 25 °C; [**8**] = 56 mM; [**17**] = 92 mM; [Acrylamide] = 321 mM; other parameters are indicated for specific experiments. **a** Experimental data showing sustained oscillations in the system that used maleimide to delay autocatalysis. [Maleimide] = 8 mM; $f/V = 1.61 \times 10^{-4}$ s$^{-1}$. **b** Experimental data showing sustained oscillations in the system that used $K_3[Fe(CN)_6]$ to delay autocatalysis. [$K_3[Fe(CN)_6]$] = 5.53 mM; $f/V = 1.34 \times 10^{-4}$ s$^{-1}$. **c** Experimental data showing the distribution of thiols during oscillations. [Maleimide] = 6.2 mM; $f/V = 1.34 \times 10^{-4}$ s$^{-1}$. **d** Phase plot reconstructed from 33 oscillatory experiments (Supplementary Section 6). Black circles denote oscillatory points; white circles denote points with damped oscillations or no oscillations; back and white circles denote special cases. The point at [Maleimide] = 6.2 mM; $f/V = 1.34 \times 10^{-4}$ s$^{-1}$ is a borderline case that demonstrates sustained oscillations in about 50% of the cases that depend on fluctuations in the experimental conditions. The points at [Maleimide] = 6.2 mM; $f/V = 2.42 \times 10^{-4}$ and $2.96 \times 10^{-4}$ s$^{-1}$ represent locally stable steady states. The red and blue colors denote the result of the linear stability analysis of the three-variable model with the following parameters: [**8**] = 56 mM; $k_1 = 0.507$ s$^{-1}$ M$^{-1}$, $k_2 = 300$ s$^{-1}$ M$^{-1}$, $k_3 = 0.0099$ s$^{-1}$, $k_4 = 3.7 \times 10^{-5}$ s$^{-1}$ (Supplementary Section 7)[60]. **e** The experimental observation of the locally stable steady state. [Maleimide] = 4 mM; $f/V = 2.96 \times 10^{-4}$ s$^{-1}$. **f** Observation of the locally stable steady state in the three-variable model. Modeling parameters: [**8**] = 56 mM; [Maleimide] = 4 mM; $f/V = 2.96 \times 10^{-4}$ s$^{-1}$; $k_1 = 0.507$ s$^{-1}$ M$^{-1}$, $k_2 = 300$ s$^{-1}$ M$^{-1}$, $k_3 = 0.0099$ s$^{-1}$, $k_4 = 3.7 \times 10^{-5}$ s$^{-1}$. Initial conditions that result in the steady state: [**8**]$_0$ = 20 mM; [Maleimide]$_0$ = $5 \times 10^{-3}$ mM; [RSH] = 2 mM. Initial conditions that result in the sustained oscillation: [**8**]$_0$ = 56 mM; [Maleimide]$_0$ = 4 mM; [RSH] = 0 mM.

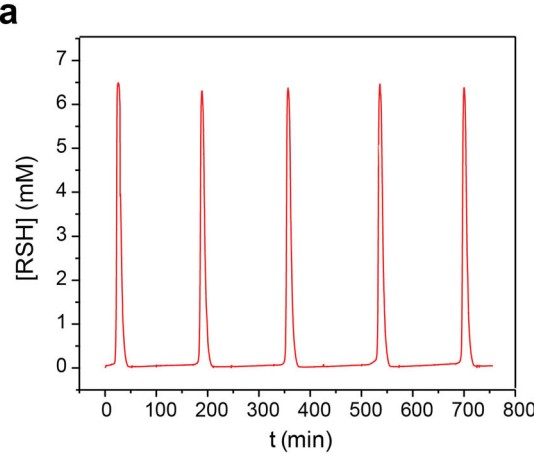

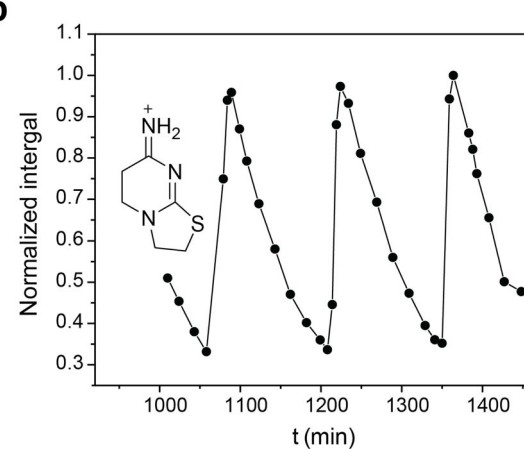

**Fig. 7 Kinetic studies of the thiouronium salt-based oscillator that uses disulfide 18.** Experimental conditions: $H_2O$, 1 M PB pH 8, at 40 °C; [**8**] = 56 mM; [**18**] = 56 mM; [Maleimide] = 8 mM; [Acrylamide] = 112 mM; $f/V = 1.61 \times 10^{-4}\,s^{-1}$. **a** Experimental data showing sustained oscillations in the concentration of thiols. **b** Experimental data showing relative changes in the amount of **22** in CSTR. Points show normalized integrals of the HPLC peak (4.4 min, SILEC Primeser 500 4.6 × 250 mm, $H_2O$ (0.1% TFA)/$CH_3CN$) corresponding to **22** obtained by analysis of the drops that come from CSTR. The detection was performed by MS detector set to $m/z = +156$.

expected, considering the excess of **17**, which is disulfide of **1**, and similar pKa values for all three thiols.

Next, we studied the robustness of the thiouronium salt-based oscillator toward variations in the flow rate and in the concentration of maleimide (Fig. 6d). To determine the region of sustained oscillations, we varied the flow rate at four concentrations of maleimide: 10, 8, 6.2, and 4 mM. Then, we correlated this information with the results of the linear stability analysis of the three-variable model (Supplementary Section 7)[40]. The results indicate the higher stability of the thiouronium salt-based oscillators than the thioester-based oscillators[39,40]. We suggest two reasons for this difference: (i) the higher ratio of the rate constants for an autocatalytic reaction to the hydrolysis reaction for **8** than for any thioester used in oscillators; and (ii) the higher rate of the reaction of thiols with acrylamide in the current oscillator than in the thioester-based oscillators that release ethanethiol. According to the model, both of these trends stabilize the oscillations.

An unexpected behavior was observed at a 4 mM concentration of maleimide. The state of the system—oscillatory or steady—depended on the initial conditions (Fig. 6e). If the $f/V$ value of $\sim 3 \times 10^{-4}\,s^{-1}$ was approached by a stepwise increase of $f/V$ from $\sim 1.6 \times 10^{-4}\,s^{-1}$, the system reached a steady state. If, however, we perturbed this steady state by flushing ($f/V$ to $6 \times 10^{-3}\,s^{-1}$) the reactor with fresh reagents for 3 min, the system settled upon sustained oscillations. This behavior is directly reproducible in the model using parameters from the experiment (Fig. 6f). Therefore, we can conclude that we observed a locally stable steady state that can be excited to the oscillatory state. Although this phenomenon is known[61,62], we were intrigued to observe it in this organic oscillator with a well-defined and simple (compare to many other chemical oscillators) mechanism.

**Oscillatory formation of the bicyclic pyrimidine derivative 22.** The disulfide **18**, which we used to study cascade cyclizations that follow autocatalysis, opened the possibility to couple oscillations to the formation of the bicyclic pyrimidine derivative **22**. To make an oscillator from **18**, we used the same setup as for the oscillator from **17** (Fig. 3), but we introduced several changes aimed at increasing the contribution of the intramolecular cyclization of **20** to the reaction network. First, we used equimolar amounts of **8** and **18** to increase the fraction of **20** in the dynamic mixture of

thiols. Second, we decreased the amount of acrylamide to decreases the competition between cyclization of **20** and its reaction with acrylamide. Third, we increased temperature, which particularly accelerates cyclizations, to 40 °C. The increase in temperature was also necessary to increase the rate of non-autocatalytic production of thiols, which dropped in comparison with the oscillator from **17**.

The oscillator demonstrated excellent stability over 12 h run (Fig. 7a). Therefore, we disconnected the flow cell for detection of thiols and began collecting drops directly from the outlet of CSTR for analysis by HPLC–MS. The analysis showed that amount of the bicycle **22** in CSTR changes periodically with an expected zigzag profile (Fig. 7b). It quickly rises because of the production from **20** during an active oscillation and then slowly drops because of washout during a lag phase.

Interestingly, the cyclization of **20** and the formation of **22** are part of the oscillatory network as additional negative feedback (Fig. 3). Thus, the generation of oscillations of **22** does not require a reaction in a separate compartment or an introduction of a parasitic pathway into the oscillatory network, but only stabilizes the oscillator.

**Increased responsiveness to environmental cues from the composite oscillator.** In the previous section, we demonstrated an oscillator that, while having a mechanism similar to the thioester oscillator, consists of different components: thiouronium salts instead of thioesters and $K_3[Fe(CN)_6]$ instead of maleimide. In the oscillatory reaction network, thiouronium salt participates in the ligation reaction with **1** as thioesters do, and $K_3[Fe(CN)_6]$ rapidly sequesters thiols as maleimide does. Nevertheless, other aspects of the chemistry of these components are very different. Thiouronium salts participate in cyclization reactions involving the amidine group, which is absent in thioesters. $K_3[Fe(CN)_6]$ reacts with reducing reagents but tolerates dienes, in contrast to maleimide.

Therefore, we designed a composite oscillator consisting of equal quantities of **8** and acetylthiocholine (**23**), $K_3[Fe(CN)_6]$, and maleimide, and that is responsive to all their substrates—furfuryl alcohol (**24**, diene), thiosemicarbazide (**25**, a reductant that can reduce $K_3[Fe(CN)_6]$ but is unable to reduce disulfide), and glyoxal (**26**, 1,2-electrophile that forms heterocycles with the amidine group)[63] (Fig. 8a, b). We compared the response of the "mixed"

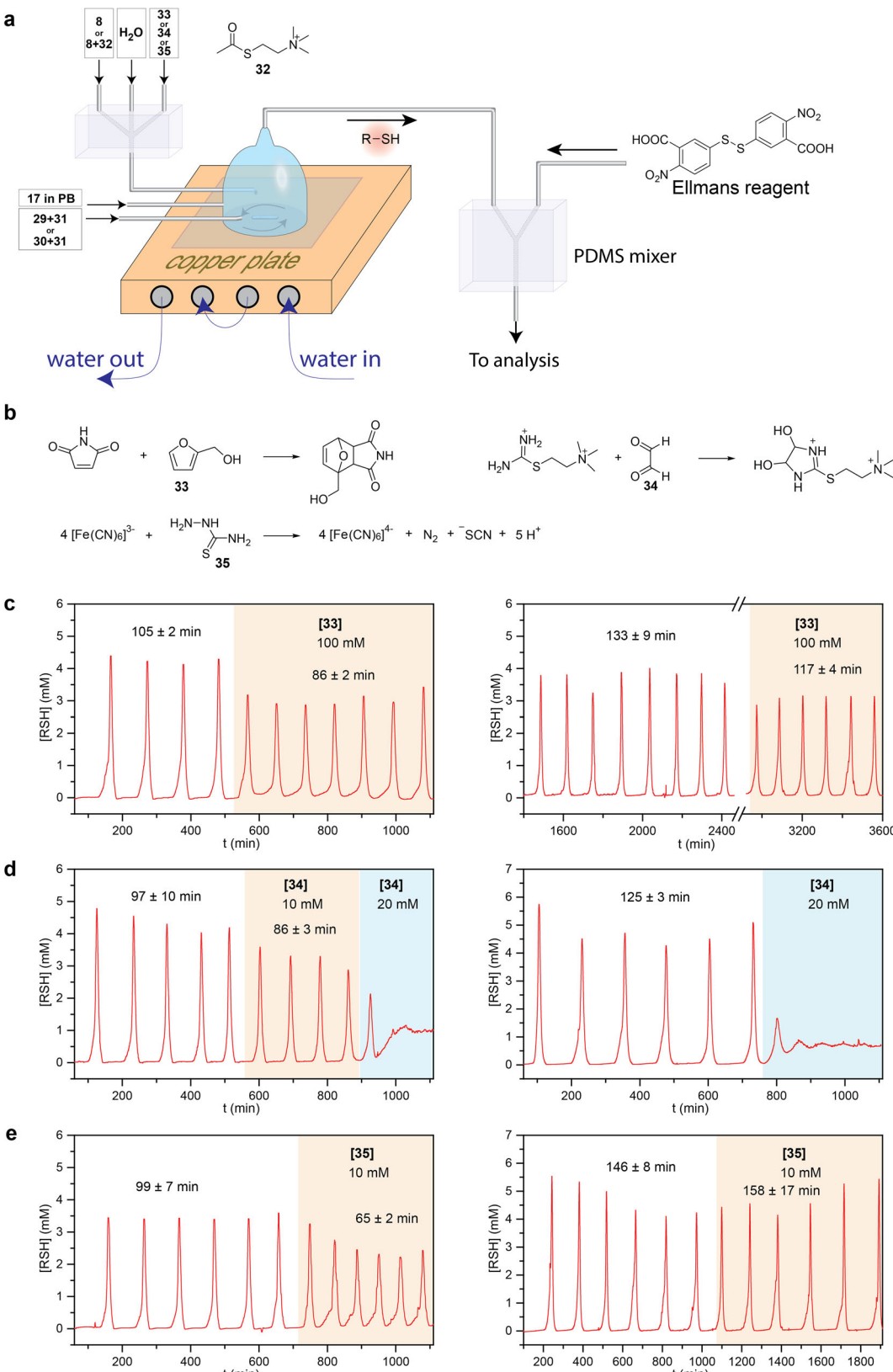

and "pure" oscillators to **33**, **34**, and **35**. Both oscillators responded to 100 mM of **33** with a shortening of the period and a decrease in the amplitude of oscillations (Fig. 8c) and to 20 mM of **34** with the disappearance of oscillations (Fig. 8d). However, only a mixed oscillator responded to 10 mM of **35** with a shortening of the period and a decrease in the amplitude of

oscillations because **35** reacts with $K_3[Fe(CN)_6]$, which had been absent in the pure system (Fig. 8e). The fact that **35** did not eliminate oscillations completely probably reflects a general trend for oscillators having two chemically orthogonal components performing the same function. These oscillators will respond to stimuli but will not lose their functionality completely because an

**Fig. 8 Response to the chemical stimuli of "pure" and "mixed" oscillators. a** A schematic representation of the CSTR experimental set-up for studying the effect of chemical stimuli on chemical oscillators. **b** The expected reactions of furfuryl alcohol (**33**), glyoxal (**34**), and thiosemicarbazide (**35**) with the components of oscillators. **c** The influence of furfuryl alcohol (100 mM) on the "mixed" (left) and "pure" (right) oscillators. In the "mixed" oscillator, the mixture of **8** (25 mM) and acetylthiocholine (**32**) (25 mM) was used instead of pure **8**, and the mixture of maleimide (5 mM) and $K_3[Fe(CN)_6]$ (5 mM) was used instead of pure maleimide. Other experimental conditions—the concentrations of **17** (92 mM) and acrylamide (321 mM), PB pH 8, at 25 °C— were as in all previous experiments; $f/V = 2.96 \times 10^{-4}\,s^{-1}$. We used the following experimental conditions in control experiments with the "pure" thiouronium salt-based oscillator: $H_2O$, 1 M PB pH 8, at 25 °C; [**8**] = 56 mM; [**17**] = 92 mM; [Acrylamide] = 321 mM; [Maleimide] = 6.2 mM; $f/V = 2.15 \times 10^{-4}\,s^{-1}$. The numbers above plots show avarage periods and their standard deviations. **d** The influence of glyoxal (10 and 20 mM) on the "mixed" (left) and "pure" (right) oscillators. **e** The influence of thiosemicarbazide (10 mM) on the "mixed" (left) and "pure" (right) oscillators.

unresponsive component (i.e., maleimide in our system) backs up the sensitive component (i.e., $K_3[Fe(CN)_6]$ in our system). By varying the ratio of responsive and unresponsive components, we should be able to control the maximum response level.

## Discussion

The importance of this work is twofold. First, the thiol-assisted formation of guanidines provides a selective method for the formation of guanidines in water at neutral pH and room temperature. These mild conditions should allow to introduce the guanidine group specifically to an amine with a neighboring thiol group in the presence of other amines.

Second, the autocatalytic and oscillatory reaction networks that we developed in this work expand our toolbox for constructing out-of-equilibrium chemical systems. Thiol-based reaction networks play a special role in this endeavor because of the high reactivity and selectivity of thiolate as nucleophile and abundance of covalent-bond exchange reactions at room temperature (e.g., thiol-disulfide and thiol-thioester exchanges) in the chemistry of thiols. Herein we link the chemistry of thiol to reactions that form guanidines and the products of their follow-up transformations. These compounds form strong hydrogen bonds, interact with polyanions[46], and are commonly involved in selective molecular recognition and self-assembly[64–66]. In this work, we have already shown how by incorporating a cascade cyclization into an oscillatory network the oscillations of a dihydropyrimidine-based heterocycle can be created. We anticipate that the autocatalytic and oscillatory reaction networks that involve close analogs of nucleic bases or that generate polyelectrolytes with guanidinium cations can now be designed and synthesized.

Native chemical ligation, which is one of the most widely used reactions in system chemistry[67–70], can be combined with the thiol-assisted formation of guanidines in one chemical system. Thus, we combined a thiouronium salt/$K_3[Fe(CN)_6]$-based oscillator with a thioester/maleimide-based oscillator into a system with increased ability to interact with external chemical stimuli. This example further highlights the benefits of complex mixtures for obtaining functional molecular systems. Cafferty et al. showed that the heterogeneity of components can increase the robustness of chemical oscillators[39]; here we showed that it increases their capability to interact with the environment.

Our studies also demonstrated how the principles outlined during the design of the thioester oscillator can be applied to design new oscillators. Importantly, we showed a complete cycle of development—from the design of a thiol-assisted ligation to the detection of chemical oscillations. In principle, an arbitrary thiol-assisted irreversible reaction could be used to build autocatalytic and oscillatory networks through this method.

## Methods

**Synthesis**. For the complete synthetic procedures of compounds 3, 4, 6–11, 16, 18, 19, 24, 26–28, *piperidine-1-carbothioamide*, *disulfide of 3-aminopropane-1-*

*thiol, 3-(2-iminothiazolidin-3-yl)propanenitrile*, we refer readers to the Supplementary Materials.

**A general protocol for the $^1H$ NMR kinetics experiments**. The kinetics of the interaction between thiouronium salts and cysteamine (**1**), its homologs **9** and **10**, or disulfides **17–19** were monitored by NMR. In a typical experiment, concentrations of all reactants were around 50 mM and reactions were performed in the phosphate buffer solution (1 M and pH 7.5 or 8). All measurements were carried out at room temperature, and changes in concentrations of the reactants and products were monitored by integration of the characteristic signals in $^1H$ NMR spectra.

**A general protocol for oscillations inflow**. A detailed description of the experimental setup used for flow experiments can be found in the Supplementary Information. In a typical experiment, four syringes were filled with the required solutions as described below. For the first syringe, 163 mg of thiouronium salt **8** were dissolved in 2.5 mL of HPLC grade water, and this solution was transferred into a syringe. An additional 0.5 mL of water was used to wash the vial in which the solution was prepared, and the volume in the solution in the syringe was precisely filled to 3 mL. The same method was used to control the total volume of the solutions when filling the other syringes. The final concentration of thiouronium salt **8** was 168 mM (in syringe) in all experiments. For the second syringe, 205 mg of acrylamide and maleimide or $K_3[Fe(CN)_6]$ in various amounts were dissolved in water. The solution was transferred to a syringe and its volume was adjusted to 3 mL. Final concentrations in the syringe were 962 mM for acrylamide and 12 mM, 18.6 mM, 24 mM, and 30 mM for maleimide or 16.6 mM for $K_3[Fe(CN)_6]$. For the third syringe, phosphate buffer (3 M) was used to prepare the solution. The buffer was prepared to have pH 8 after a threefold dilution to a concentration of 1 M. Thus, to obtain 100 mL of a buffer solution, 4.325 g (0.0318 mol) of anhydrous $KH_2PO_4$ and 46.347 g (0.2679 mol) of anhydrous $K_2HPO_4$ were placed in a 100 mL volumetric flask and filled with HPLC grade water up to 100 mL. This PBS was used to prepare a 3 mL solution of 186 mg of cystamine dihydrochloride salt in the syringe. The final concentration of cystamine in the syringe was 276 mM for all experiments. For the last syringe, two solutions were prepared separately. First, 10 mL of HPLC grade methanol was used to dissolve 164 mg (0.414 mmoles) of 5,5′-dithiobis-(2-nitrobenzoic acid) (Ellman's reagent) and 15 mL of HPLC grade water was used to dissolve 0.5 g of $KH_2PO_4$. Those two solutions were combined and a 25 mL glass syringe was filled with a solution through a 0.22 μm syringe filter. The final concentration of Ellman's reagent in the syringe was 16.6 mM.

All 5 mL syringes (with 3 mL of solution each) were installed in the syringe pump system and connected to the CSTR using 0.5 mm internal diameter PTFE tubing. The CSTR outlet tubing was connected to the microfluidic mixer where the content of CSTR was mixed with Ellman's reagent. In all experiments, the flow of the solution of Ellman's reagent was three times higher than the flow from CSTR. After mixing, the flow passed through the flow cell, where absorbance at 412 nm was measured.

Experiments with the oscillator based on the disulfide **18** were performed in the same way, but the composition of syringes was changed because of the phase separation of the deprotonated **18** in 3 M phosphate buffer. Thus, the content of the syringes was as following: syringe 1 (**18** in form of dihydrochloride, 168 mM); syringe 2 (phosphate buffer, $K^+$, 3 M, pH 8); syringe 3 (**8** 168 mM, maleimide 24 mM, acrylamide 336 mM). No modifications were made to the syringe with Ellman's reagent and the detection system. To analyze oscillation is bicycle **22**, an individual drops (~15 μL each) of solution were collected directly from the outlet of CSTR and analyzed by HPLC–MS (SILEC Primeser 500 4.6 × 250 mm, $H_2O$ (0.1% TFA)/$CH_3CN$).

## Data availability

The authors declare that all data supporting the findings of this study are available within the paper and its supplementary information files. The X-ray crystallographic coordinates for structures reported in this study have been deposited at the Cambridge Crystallographic Data Centre (CCDC), under deposition numbers 2033291, 2063063, 2062125, and 2042595. These data can be obtained free of charge from The Cambridge

Crystallographic Data Centre via https://www.ccdc.cam.ac.uk/data_request/cifwww.ccdc.cam.ac.uk/data_request/cif.

## Code availability

The MATLAB code used in this paper is available from https://doi.org/10.5281/zenodo.4605227.

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

## Acknowledgements

This work was supported by the Israel Science Foundation (grant 2333/19, to S.N.S.) and by a research grant from the Weizmann SABRA—Yeda-Sela—WRC Program, the Estate of Emile Mimran, and The Maurice and Vivienne Wohl Biology Endowment. A.I.N. thanks Israel Ministry of Absorption for financial support.

## Author contributions

S.N.S. supervised the research. S.N.S., A.I.N., and A.I.H. planned the project and designed the experiments. A.I.N., A.I.H., X.M., and S.N.S. performed the experiments and analyzed the data. A.I.N., X.M., and S.N.S. performed the numerical simulations. L.J.W.S. and Y.D.-P. performed the crystal X-ray analysis. S.N.S. wrote the paper with help from A.I.N., A.I.H., and X.M.

## Competing interests

The authors declare no competing interests.
