## [Peer Review File · Nature Communications]

Reviewer #1 (Remarks to the Author):

Cyclic chemical processes arise naturally and play an important role in the transformation of chemical energy into work in soft matter systems. There are indeed limited examples of synthetic organic oscillators. Here the authors introduce a new chemical oscillator involving thiouronium salts and formation of guanidines. This is a really beautiful piece of work, carefully performed and well characterized. It is certainly novel and this oscillator may have interesting applications. I believe this manuscript should be published.

Line 174 It's not clear to me why use addition of mercaptoethanol to eliminate the induction period, why not add the autocatalyst 1?

Line 226, I would probably say residence time rather than species lifetime which is a term often associated with the kinetics.

Figure 5 has a particularly long caption, might be clearer if some of the detail was removed if it is in the main text eg how the data is obtained.

Line 310. Systems with only three variables can show all kinds of interesting behavior, including chaos. This might be a subcritical Hopf, usually difficult to stabilize experimentally. Then it is a testament to the carefully controlled experimental set-up. Perhaps better to say you were surprised to observe it in this new organic system or something similar.

Reviewer #2 (Remarks to the Author):

The manuscript of Semenov and coworkers, entitled "Autocatalytic and Oscillatory Reactions that Form Substituted Guanidines" presents very interesting new versions or more precisely a new subclass of the previously published thiol autocatalysis based organic oscillators (Semenov, S., Kraft, L., Ainla, A. et al. Autocatalytic, bistable, oscillatory networks of biologically relevant organic reactions. *Nature* 537, 656–660 (2016).). The manuscript is well written, the figures help the reader to understand the chemistry and to follow the experimental procedures. The applied methodology is appropriate, the use of CSTR is a standard tool to get chemical oscillations. The description of the experiments is detailed enough to reproduce the results.

Questions and comments:

1) Please point out the main differences of these new systems and the previously published thiol autocatalysis based organic oscillators (Semenov, S., Kraft, L., Ainla, A. et al. Autocatalytic, bistable, oscillatory networks of biologically relevant organic reactions. *Nature* 537, 656–660 (2016).) Does the mechanism of the autocatalysis change with the replacement of the previously used L-alanine ethyl thioester by thiouronium salts? I find the mechanism presented in Figure 1 in the previous paper (*Nature* 537, 656–660) quite similar to the one in this manuscript in Figure 3. Do these systems belong to the same class or are they inherently different?

2) The design presented here follows the design method, which has theoretically confirmed (e.g. [J. Boissonade and P. De Kepper, Transitions from bistability to limit cycle oscillations. Theoretical analysis and experimental evidence in an open chemical system, *J. Phys. Chem.*, 84 (1980), 501–506.] and [G. Dangelmayr and J. Guckenheimer, On a four parameter family of planar vector fields, *Archive for Rational Mechanics and Analysis*, 97 (1987), 321–352.]) and has been successfully used

since the '80s to design a various class of chemical oscillators (I. R. Epstein, K. Kustin, P. De Kepper and M. Orbán, "Oscillating Chemical Reactions," *Sci. Amer.* 248(3), 112-123 (1983).). Not only the experimental strategy but the suggested model and simulations presented in the supporting material shows directly this connection. My opinion is, that this fact must be mentioned in the introduction.

This design method is based on the fact that a bistable (autocatalytic) system can be readily changed into an oscillatory system by the addition of an appropriate feedback reaction. Now, bistability is a common phenomenon when an autocatalytic reaction is performed in a CSTR. Do the authors observe bistability between the stationary states of the CSTR content?

In the light of the above-mentioned theoretical works the observed bistability between an oscillatory state and a steady-state is not surprising, but the consequence of the general dynamics of these types of networks. For instance, this can be the sign of a subcritical Hopf-bifurcation. This phenomenon is well know and has been observed in different inorganic oscillators, see e.g. M. Orban Oscillations and bistability in the copper(II)-catalyzed reaction between hydrogen peroxide and potassium thiocyanate *J. Am. Chem. Soc.* 1986, 108, 22, 6893–6898.

3) In the introduction the authors state, that "Thioesters-based oscillators remain the only example of purely organic oscillators". Later on the work of th A. Taylor group, entitled "An Organic-Based pH Oscillator" is cited. Can we consider the systems of base-catalyzed dehydration of a hydrated carbonyl compound as organic oscillators or not? There are also reports on oscillations during the Maillard reaction, e.g. Voeikov, V.L., Koldunov, V.V. & Kononov, D.S. *New Oscillatory Process in Aqueous Solutions of Compounds Containing Carbonyl and Amino Groups*1. *Kinetics and Catalysis* 42, 606–608 (2001). I think these systems are also organic reaction-based chemical oscillators. If the authors agree, than introduction should be refined.

4) I agree with the statement of the authors, that the inorganic chemistry based oscillators and the biochemical oscillators are quite different. But, I do not think, that the main point is the strong oxidizing nature or the extreme pH of the inorganic systems (in fact it is not always the case). More significant is the difference in the mechanism of the oscillations, which have been pointed out by Novák and Tyson (*Design principles of biochemical oscillators. Nat Rev Mol Cell Biol* 9, 981–991). The dominant mechanism of inorganic chemistry based oscillators is direct autocatalysis (nonlinear positive feedback). However, this is rarely the case in biochemistry. In biochemistry oscillations are typically based on delayed negative feedback loops. Considering it, I would say the mechanism of these organic oscillators are much closer (or even the same) to the one of the inorganic oscillators, since it is based on direct autocatalysis (see the supporting file $S+A \rightarrow 2A$). I would like to ask the authors to consider and analyze this mechanistic aspect also in their paper.

5) I would suggest not to call a single cycle (a pulse) to oscillations like in "S4.3. Batch experiments showing single oscillation".

Finally, I want to mention again that the experimental work is quite impressive and this work is an important development in the field of oscillatory chemical reactions.

Reviewer #3 (Remarks to the Author):

The work by Semenov et al. describes the development of autocatalytic and oscillating systems by rational design of chemical reactions. The manuscript is well written, the study is well done, and the topic is of interest to a broad readership, including life-like materials and origins of life. However, the level of novelty of this work is questionable as it is a variation of references [39] and [40] of the last author. Compared to these publications, the innovations are:

- A. replacing thioesters by thiouronium salts as feedstock of the autocatalytic reaction;
- B. “demonstrate the possibility of conversion of guanidines formed in the autocatalytic reaction into analogs of nucleobases”;
- C. replacing maleimide by $K_3[Fe(CN)_6]$ as an inhibitor for oscillations, and play with the various chemicals to perturb the dynamics of the oscillators.

Concerning point A, the expansion was implemented through minor structural changes (amide-to-amidine, O-to NH substitution). Note that modification of the feedstock is done in reference [39], but with other types of thioesters only. The reaction between S-substituted amidines and amines that yield guanidines is not surprising. Such bioconjugation applications in itself pertains to a more specialized journal. This point however enables point B.

Concerning point B, guanidine is undoubtedly a more versatile precursor for further functionalization. A first problem is that the example of 2-amino-6-methylpyrimidin-4(3H) formation is not sufficient to fully appreciate the functional potential of this building block. A second problem is that this reaction is done separately from the autocatalytic reaction (chemical ‘18’ is resynthesized) and there is no demonstration of both occurring together. In this sense, the claim of the authors that they “demonstrate the possibility...” is misleading. What is shown at this stage is rather a preliminary study toward showing autocatalysis coupled to reactions yielding products of interest.

Concerning point C, if the tunability of the system by external cues is to be the main claim of the paper, the changes in amplitude and frequency should be quantified (these changes are only visually displayed and left to the judgement of the reader) and such quantification should be done for a range of conditions that is systematic enough to appreciate the generality and limits of modulations (e.g. similarly to what the authors do in Fig. 5d for the existence of oscillations, or in Fig. 3c of their reference [39]).

In conclusion, the authors demonstrate a list of improvements that are promising for future engineering of such networks but it is unclear whether any of these improvement or their sum is sufficient for publication in Nature Communications. A clear-cut element of novelty would have been to actually realize any of what is proposed in the last phrase of the introduction (“Having guanidines as an output of the oscillator opens possibilities to couple the oscillator’s output to the formation of polyelectrolyte assemblies...”), but this is not done in the present work.

Minor comment:

P8: "The graph shows the expected negative correlation between the pKa of a leaving group and the rate of nucleophilic substitution": replace 'correlation' by 'trend', correlation is a statistical quantity.

Referees' comments:

Referee #1 (Remarks to the Author):

Cyclic chemical processes arise naturally and play an important role in the transformation of chemical energy into work in soft matter systems. There are indeed limited examples of synthetic organic oscillators. Here the authors introduce a new chemical oscillator involving thiouronium salts and formation of guanidines. This is a really beautiful piece of work, carefully performed and well characterized. It is certainly novel and this oscillator may have interesting applications. I believe this manuscript should be published

Our Response: We thank the reviewer for his/her description of the manuscript. Oscillators indeed play a very important role in living systems, but very difficult to design *de novo* without use of enzymes or RNA/DNA.

Line 174 It's not clear to me why use addition of mercaptoethanol to eliminate the induction period, why not add the autocatalyst 1

Our Response: We agree that use of autocatalyst 1 (cysteamine) is the most direct approach to demonstrate elimination of the lag phase. We, however, notice that almost any added thiol, including mercaptoethanol, will quickly undergo disulfide exchange with cysteamine (compound 17) and produce autocatalyst 1. The mercaptoethanol was used out of practical convenience. We therefore opted to keep this experiment as is.

Line 226, I would probably say residence time rather than species lifetime which is a term often associated with the kinetics.

Our Response:

We completely agree with the referee that residence time is more precise and appropriate term than life in this context. We changed lifetime to residence time in the text.

P 16

Text removed: lifetime.

Text added: residence time.

Figure 5 has a particularly long caption, might be clearer if some of the detail was removed if it is in the main text eg how the data is obtained.

Our Response: We agree with referee. The caption for figure 5 was too long. We shortened it by removing the repetition of the identical part of the experimental conditions and by moving some descriptions of experiments to the manuscript's main text. In addition, we corrected two typos in this caption: (i) k_3 in simulations was changed from 0.033 to 0.0099; (ii) k_4 in f was changed from $3.8 \cdot 10^{-5}$ to $3.7 \cdot 10^{-5}$. No calculations or plots were changed.

Caption 6

Text removed: Experimental conditions: H₂O, 1 M PB pH 8, at 25 °C; [8] = 56 mM; [17] = 92 mM; [Acrylamide] = 321 mM;

Experimental conditions: H₂O, 1 M PB pH 8, at 25 °C; [8] = 56 mM; [17] = 92 mM; [Acrylamide] = 321 mM;

The data are based on an HPLC analysis of assymetrical disulfides formed in reactions with Ellman's reagent. Experimental conditions: H₂O, 1 M PB pH 8, at 25 °C; [8] = 56 mM; [17] = 92 mM; [Acrylamide] = 321 mM;

Experimental conditions: H₂O, 1 M PB pH 8, at 25 °C; [8] = 56 mM; [17] = 92 mM; [Acrylamide] = 321 mM;

To perturb this steady state and initiate oscillations, we increased f/V to $6 \cdot 10^{-3} \text{ s}^{-1}$ for 3 minutes.

Text added: . Experimental conditions shared by all experiments: H₂O, 1 M PB pH 8, at 25 °C; [8] = 56 mM; [17] = 92 mM; [Acrylamide] = 321 mM; other parameters are indicated for specific experiments.

P 19-20

Text added: which clearly resolved assymetrical disulfides formed in reactions of **1**, **11**, and thiocholine with Ellman's reagent.

If, however, we perturbed this steady state by flushing (f/V to $6 \cdot 10^{-3} \text{ s}^{-1}$) the reactor with fresh reagents for 3 minutes, the system settled upon sustained oscillations.

Line 310. Systems with only three variables can show all kinds of interesting behavior, including chaos. This might be a subcritical Hopf, usually difficult to stabilize experimentally. Then it is a testament to the carefully controlled experimental set-up. Perhaps better to say you were surprised to observe it in this new organic system or something similar.

Our Response: We thank the referee for pointing that some systems with three variables are capable for complex behaviors. We corrected the text according to referee's suggestion. We also looked closely into the stability of the steady states at the parameters resulting in bistability between the steady and oscillatory states in the numerical model (see Figure). The analysis shows that the system undergoes neither subcritical nor supercritical Hopf bifurcation. The "high thiol concentration" steady state does not lose its local stability until it collides with the unstable steady state and disappears (probably through saddle-node bifurcation). Therefore, the birth and death of the stable orbit, which we observed in the numerical simulations, belongs to one of the global bifurcations of cycles.

Response to Referees Figure 1. The plot of steady states and their stability. Red color indicates unstable state; cyan stable. Modeling parameters: $[8] = 56 \text{ mM}$; $[\text{Maleimide}] = 4 \text{ mM}$; $k_1 = 0.507 \text{ s}^{-1}\text{M}^{-1}$, $k_2 = 300 \text{ s}^{-1}\text{M}^{-1}$, $k_3 = 0.0099 \text{ s}^{-1}$, $k_4 = 3.7 \cdot 10^{-5} \text{ s}^{-1}$.

P 20

Text removed: system with only three variables.

Text added: Although this phenomenon is known,⁵⁵ we were intrigued to observe it in this organic oscillator with well-defined and simple (compare to many other chemical oscillators) mechanism.

Referee #2 (Remarks to the Author):

The manuscript of Semenov and coworkers, entitled "Autocatalytic and Oscillatory Reactions that Form Substituted Guanidines" presents very interesting new versions or more precisely a new subclass of the previously published thiol autocatalysis based organic oscillators (Semenov, S., Kraft, L., Ainla, A. et al. Autocatalytic, bistable, oscillatory networks of biologically relevant organic reactions. Nature 537, 656–660 (2016).). The manuscript is well

written, the figures help the reader to understand the chemistry and to follow the experimental procedures. The applied methodology is appropriate, the use of CSTR is a standard tool to get chemical oscillations. The description of the experiments is detailed enough to reproduce the results..

Our Response:

We thank the reviewer for his/her remarks. Indeed, one point of the manuscript is that it demonstrates how mechanistic principles outlined in Nature 2016 can be used to design subclasses of chemical oscillators with new chemistry.

1) Please point out the main differences of these new systems and the previously published thiol autocatalysis based organic oscillators (Semenov, S., Kraft, L., Ainla, A. et al. Autocatalytic, bistable, oscillatory networks of biologically relevant organic reactions. Nature 537, 656–660 (2016).) Does the mechanism of the autocatalysis change with the replacement of the previously used L-alanine ethyl thioester by thiouronium salts? I find the mechanism presented in Figure 1 in the previous paper (Nature 537, 656–660) quite similar to the one in this manuscript in Figure 3. Do these systems belong to the same class or are they inherently different?

Our Response: The *mechanisms* of these oscillators is indeed the same. The reactions that are the thermodynamic driving force for these reactions (and oscillations) are different. The oscillator presented in this paper is driven by thiol-assisted formation of guanidines, which was also firstly demonstrated in this paper. As result, the products that these oscillators produce belong to different classes of compounds: guanidines (this paper) vs amides (in 2016 paper). During the revision, to further highlight difference between guanidines and amides, we designed a cascade reaction where formation of guanidines is followed up with cyclization leading first to dihydropyrimidines and later to pyrimidines. These types of cascade cyclization are impossible for amides.

2) The design presented here follows the design method, which has theoretically confirmed (e.g. [J. Boissonade and P. De Kepper, Transitions from bistability to limit cycle oscillations.

Theoretical analysis and experimental evidence in an open chemical system, J. Phys. Chem., 84 (1980), 501–506.] and [G. Dangelmayr and J. Guckenheimer, On a four parameter family of planar vector fields, Archive for Rational Mechanics and Analysis, 97 (1987), 321–352.] and has been successfully used since the '80s to design a various class of chemical oscillators (I. R. Epstein, K. Kustin, P. De Kepper and M. Orbán, "Oscillating Chemical Reactions," Sci. Amer. 248(3), 112-123 (1983).). Not only the experimental strategy but the suggested model and simulations presented in the supporting material shows directly this connection. My opinion is, that this fact must be mentioned in the introduction.

Our Response: We agree that this oscillator belongs to the class of nonlinear chemical systems that display cross-shaped diagram described in (J. Phys. Chem., 84 (1980), 501–506) and can be designed from bistable chemical systems. We, however, felt that this discussion would not fit well into current introduction; therefore we discussed these aspects of the oscillator in the discussion section.

Changes to the text:

P14.

Text added: The design approach presented here also reassembles the strategy that was theoretically outlined by Boissonade and De Kepper and was experimentally used by Epstein and colleagues.^{32, 54, 55} The method states that a bistable chemical system can be converted into oscillatory by the addition of an appropriate negative feedback species. For the oscillator in this manuscript, the negative feedback species is acrylamide. The bistable system is thiouronium salt-based autocatalytic network combined with maleimide or $K_3[Fe(CN)_6]$.

This design method is based on the fact that a bistable (autocatalytic) system can be readily changed into an oscillatory system by the addition of an appropriate feedback reaction. Now, bistability is a common phenomenon when an autocatalytic reaction is performed in a CSTR. Do the authors observe bistability between the stationary states of the CSTR content?

Our Response: Since thiouronium salts based autocatalytic network represent a quadratic autocatalysis, it requires addition of maleimide or $K_3[Fe(CN)_6]$ to display bistability in CSTR.

We performed an experiment (see Figure below) which confirms that the system is capable of bistable behavior. We, however, decided to not study bistability in this system in details and include it in the manuscript because it will significantly increase the length of this already lengthy manuscript, while fundamental nature of this bistability is similar to the bistability in Taylor's system and in thioester based system.

Response to Referees Figure 2. An experimental data showing bistable behavior. Experimental conditions: H₂O, 1 M PB pH 8; [**8**] = 56 mM; [**17**] = 92 mM; [Maleimide] = 8 mM. Absorbance values were not converted to concentration of thiols because, in contrast to oscillatory experiments, thiol concentration reaches values beyond the detection window of our flow cell.

In the light of the above-mentioned theoretical works the observed bistability between an oscillatory state and a steady-state is not surprising, but the consequence of the general dynamics of these types of networks. For instance, this can be the sign of a subcritical Hopf-bifurcation. This phenomenon is well known and has been observed in different inorganic oscillators, see e.g. M. Orban Oscillations and bistability in the copper(II)-catalyzed reaction

between hydrogen peroxide and potassium thiocyanate *J. Am. Chem. Soc.* 1986, 108, 22, 6893–6898.

Our Response: In response to this comment and also to the related comment by referee 1, we changed the sentence discussing bistability between the steady and oscillatory states and included the reference mentioned by the referee.

We also looked closely into the stability of the steady states at the parameters resulting in bistability between the steady and oscillatory states in the numerical model (see Figure). The analysis shows that the system undergoes neither subcritical nor supercritical Hopf bifurcation. The “high thiol concentration” steady state does not lose its local stability until it collides with the unstable steady state and disappears (probably through saddle-node bifurcation). Therefore, the birth and death of the stable orbit, which we observed in the numerical simulations and experiments, belongs to one of the global bifurcations of cycles.

Response to Referees Figure 1 (Copy). The plot of steady states and their stability. Red color indicates unstable state; cyan stable. Modeling parameters: $[8] = 56 \text{ mM}$; $[\text{Maleimide}] = 4 \text{ mM}$; $k_1 = 0.507 \text{ s}^{-1}\text{M}^{-1}$, $k_2 = 300 \text{ s}^{-1}\text{M}^{-1}$, $k_3 = 0.0099 \text{ s}^{-1}$, $k_4 = 3.7 \cdot 10^{-5} \text{ s}^{-1}$.

Changes to the text:

P 20

Text removed: system with only three variables.

Text added: Although this phenomenon is known,⁵⁵ we were intrigued to observe it in this organic oscillator with well-defined and simple (compare to many other chemical oscillators) mechanism.

3) In the introduction the authors state, that "Thioesters-based oscillators remain the only example of purely organic oscillators". Later on the work of th A. Taylor group, entitled "An Organic-Based pH Oscillator" is cited. Can we consider the systems of base-catalyzed dehydration of a hydrated carbonyl compound as organic oscillators or not? There are also reports on oscillations during the Maillard reaction, e.g. Voeikov, V.L., Koldunov, V.V. & Kononov, D.S. New Oscillatory Process in Aqueous Solutions of Compounds Containing Carbonyl and Amino Groups1. Kinetics and Catalysis 42, 606–608 (2001). I think these systems are also organic reaction-based chemical oscillators. If the authors agree, than introduction should be refined.

Our Response. We agree that the definition of an “organic” oscillator is somewhat ambiguous. The introduction mainly focused on the structural flexibility of organic molecules. Although formaldehyde is an organic molecule, its structure cannot be modified without a dramatic change in chemical reactivity (also, the autocatalyst in this oscillator is OH⁻, which is clearly an inorganic particle). Clearly, the oscillator from the Taylor group can be considered organic, but it would not fit into the context of the introduction (as the referee mentioned, the work is cited in the discussion section of the manuscript). Therefore, we modified the sentence in the introduction to make it more specific and to avoid the term “organic oscillator”. Regarding the example of the Maillard reaction, it involves gaseous oxygen, and gradients of oxygen concentration in solution were reported by authors. Thus, it cannot be considered a homogenous oscillator. Several other reactions of oxidation of organic compounds by O₂ display oscillations (e.g. oxidation of benzaldehyde J. Phys. Chem. 1990, 94, 4404). We however opted to limit discussions to homogenous oscillators.

Changes to the text

P 3

Text removed: Thioesters-based oscillators remain the only example of purely organic oscillators.

Text added: Thioesters-based oscillators remain the only example of homogenous oscillators where all reactants and products are organic molecules.^{39, 40}

4) I agree with the statement of the authors, that the inorganic chemistry based oscillators and the biochemical oscillators are quite different. But, I do not think, that the main point is the strong oxidizing nature or the extreme pH of the inorganic systems (in fact it is not always the case). More significant is the difference in the mechanism of the oscillations, which have been pointed out by Novák and Tyson (Design principles of biochemical oscillators. Nat Rev Mol Cell Biol 9, 981–991). The dominant mechanism of inorganic chemistry based oscillators is direct autocatalysis (nonlinear positive feedback). However, this is rarely the case in biochemistry. In biochemistry oscillations are typically based on delayed negative feedback loops. Considering it, I would say the mechanism of these organic oscillators are much closer (or even the same) to the one of the inorganic oscillators, since it is based on direct autocatalysis (see the supporting file S+A->2A). I would like to ask the authors to consider and analyze this mechanistic aspect also in their paper.

Our Response.

We agree with the referee that biochemical oscillators usually involve delayed negative feedback. Actually, the synthetic biochemical oscillators (e.g. Nat. Chem. 2015, 7, 160; Molecular systems biology 2011, 7, 466; Molecular systems biology 2011, 7, 465) often combine autocatalysis (positive feedback) with delayed negative feedback. One of the key aspects of the design of this oscillator (and also of the oscillator in Nature 2016) is that the design does not start from a an experimentally found bistable system. It is based on quadratic autocatalysis generated by autocatalytic reaction network shown in figure 3. Because of the quadratic nature of the autocatalysis, it is incapable of bistable behavior in flow conditions. Therefore, the design involves addition of maleimide or $K_3[Fe(CN)_6]$ that sequester thiols formed in hydrolysis reaction, which is not a part of the autocatalytic network, to make the system bistable. We reflected the point about lacking delayed negative feedback in the sentence shown below.

Changes to the text:

P17.

Text added: Therefore, the substrate depletion and the trigger are the main sources of instability in this oscillator in contrast to the majority of biological oscillators where delayed negative feedback is the source of instability.⁵⁷

5) I would suggest not to call a single cycle (a pulse) to oscillations like in "S4.3. Batch experiments showing single oscillation"

Our Response. We changes the term “single oscillation” to “pulse” thought the text.

Referee #3 (Remarks to the Author):

The work by Semenov et al. describes the development of autocatalytic and oscillating systems by rational design of chemical reactions. The manuscript is well written, the study is well done, and the topic is of interest to a broad readership, including life-like materials and origins of life. However, the level of novelty of this work is questionable as it is a variation of references [39] and [40] of the last author. Compared to these publications, the innovations are:

A. replacing thioesters by thiouronium salts as feedstock of the autocatalytic reaction;

B. “demonstrate the possibility of conversion of guanidines formed in the autocatalytic reaction into analogs of nucleobases”;

C. replacing maleimide by $K_3[Fe(CN)_6]$ as an inhibitor for oscillations, and play with the various chemicals to perturb the dynamics of the oscillators.

Our Response: We appreciate the referee’s concern about novelty. In response, we added a significant amount of the experimental work that demonstrates the uniqueness of guanidine chemistry in autocatalytic and oscillatory reactions through the cascade cyclizations, one-pot formation of analogs of nucleobases, and coupling of oscillation to the formation of a

dihydropyrimidine based heterocycle. The details of this work are presented in the response to the specific concerns.

Concerning point A, the expansion was implemented through minor structural changes (amide-to-amidine, O-to NH substitution). Note that modification of the feedstock is done in reference [39], but with other types of thioesters only. The reaction between S-substituted amidines and amines that yield guanidines is not surprising. Such bioconjugation applications in itself pertains to a more specialized journal. This point however enables point B.

Our Response:

O to NH substitution looks small on paper but in most cases has dramatic chemistry consequences; it is enough to look to alcohols and amines (ROH vs RNH₂) or esters and amides (RCOOR' vs RCONHR'). In the present example, it was reasonable to hypothesize that thiouronium salts, which differ from thioesters in two positions O to NH change and R- to RNH- change, would be susceptible to thiol-assisted ligation, however, this conclusion cannot be directly deduced from literature data and general chemistry principles without experimental work for the following reasons. For thiol-assisted ligation to work, both steps, thiol exchange and intramolecular rearrangement, have to have high rates in water at neutral pH. Although the literature data supporting the hypothesis about fast intramolecular rearrangement exist (J. Am. Chem. Soc. (1957) 79, 5663-5666), we found no kinetic data on thiouronium salts thiol exchange. In fact, thiol exchange often a limiting step even in native chemical ligation (J. Am. Chem. Soc. 2006, 128, 20, 6640). Moreover, our preliminary results indicate that even smaller change when switching from thioesters (RCOSR) to thiocarbonates (ROCOSR) causes inhibition of thiol exchange to the level where thiol-assisted ligation is inefficient.

As the referee mentioned, an important consequence of forming guanidines is that it enables subsequent formation of various heterocycles. The point about the formation of heterocycles was strongly strengthened during this revision as described in the response to the comment below.

Concerning point B, guanidine is undoubtedly a more versatile precursor for further functionalization. A first problem is that the example of 2-amino-6-methylpyrimidin-4(3H)

formation is not sufficient to fully appreciate the functional potential of this building block. A second problem is that this reaction is done separately from the autocatalytic reaction (chemical '18' is resynthesized) and there is no demonstration of both occurring together. In this sense, the claim of the authors that they "demonstrate the possibility..." is misleading. What is shown at this stage is rather a preliminary study toward showing autocatalysis coupled to reactions yielding products of interest.

Our Response: We agree with the referee that further work was desirable to highlight the uniqueness of the chemistry of guanidines for reaction networks. Following experimental results were added:

1. We synthesized two new disulfides bearing nitrile groups: $(\text{NCCH}_2\text{CH}_2\text{NHCH}_2(\text{CH}_2)_n\text{S})_2$ $n = 1$ and 2 .
2. We showed that in reaction with thiuronium salts these disulfides first autocatalytically give corresponding guanidines, which spontaneously and in the same reaction mixture transform into bicyclic derivatives of dihydropyrimidine (new Fig. 4a-c)
3. We showed that some of these derivatives of dihydropyrimidine can be converted into derivatives of pyrimidine by the one-pot procedure that consists of drying of the products of the autocatalytic step and heating the solid residue with iodine (new Fig. 4a).
4. We also showed that a derivative of aminopyrimidine can be obtained in a one-pot procedure consisting of the autocatalytic formation of 2-mercaptoethylguanidine, followed by its cyclization into aminothiazolidine at 80°C and by addition of malononitrile and pyridine (new Fig.4d, f).

Changes to the text:

A new figure 4 was added. In addition to the illustrations for the experiments described above, the panels d and e from figure 3 were moved to the figure 4.

Main text Figure 4. Cyclization of guanidines formed in autocatalytic reactions. **a.** Reactions of disulfides **18** and **19** with **4**. Compounds **20-23** form spontaneously in water at pH 8. Compounds **24-25** form in the one-pot process. **b.** An ORTEP diagram for X-ray structure of **24** (50% probability ellipsoids). **c.** The kinetics of the reactions of **18** and **19** with **4**. Concentrations of **20** and **22** were monitored in the reaction of **18** by ^1H NMR. The concentration of **21** was monitored in a separate reaction with **19**. Reactions conditions: D_2O , 1 M PB pH 8, 25°C , $[\mathbf{4}] = 70\text{ mM}$; $[\mathbf{18}$ or $\mathbf{19}] = 70\text{ mM}$). **d.** One-pot formation of the aminopyrimidine **26** from **4**, **17**, and malononitrile. **e.** Scheme of the reaction between disulfide **18** and ethyl acetoacetate. **f.** An

ORTEP diagram for X-ray structure of **26** (50% probability ellipsoids). **g.** An ORTEP diagram for X-ray structure of **28** (50% probability ellipsoids). Hydrogen bonds are denoted as blue dashed lines.

P11

Text added: **Cyclization of the guanidines formed in the autocatalytic reaction.** One of the major chemical differences between amides and derivatives of guanidines is that the former relatively easily participate in various cyclization reactions that give heterocycles including analogs of nucleobases. To demonstrate cascade cyclizations of the autocatalytically formed guanidines, we synthesized disulfides **18** and **19** by reacting correspondingly **17** and the disulfide of 3-aminopropane-1-thiol with acrylonitrile. Molecules **18** and **19** undergo autocatalytic reaction with thiouronium salt **4** (Fig. 4a-b). The autocatalytic profile of the reactions of **18** with **4** is similar to the profile of the reaction of **17** and **4** indicating that transition from primary to secondary amine has little influence on the ligation rate (Fig. 4c). The guanidines **20** and **21** further undergo cascade cyclization leading to bicycles **22** and **23** that contain dihydropyrimidine ring. Expectedly, the formation of **22** is much faster than the formation of **23**. The identity of the compounds **22-23** in the reaction mixture was confirmed by HPLC-MS analysis and, for **22**, by comparing signals in ¹H NMR of the reaction mixture with signals of the cyclization product of 3-(2-iminothiazolidin-3-yl)propanenitrile (Supplementary section 5). Evaporation of the reaction mixture resulted in the partial hydrolysis of **22** to **24**, which was isolated by chromatography. The identity of **24** was confirmed by NMR and X-ray crystallography (Fig. 4b). Heating the slurry left after evaporation of the reaction mixture with I₂ provided the pyrimidine derivative **25** which shows two characteristic doublets in ¹H NMR at 7.81 and 5.83 ppm.

Iminothiazolidine, which forms from **11** upon heating, was another interesting target for the one-pot formation of heterocycles.^{54, 55} Thus, heating the products of the autocatalytic reaction between

4 and **17** with the subsequent addition of pyridine and malononitrile results in the formation of the aminopyrimidine derivative **26**, which was identified by comparison with a separately synthesized standard (Fig. 4d, f and Supplementary section 5)

Supplementary section 5 that describes the analysis of the heterocycles discussed in the main text was also added.

Concerning point C, if the tunability of the system by external cues is to be the main claim of the paper, the changes in amplitude and frequency should be quantified (these changes are only visually displayed and left to the judgement of the reader) and such quantification should be done for a range of conditions that is systematic enough to appreciate the generality and limits of modulations (e.g. similarly to what the authors do in Fig. 5d for the existence of oscillations, or in Fig. 3c of their reference [39]).

Our Response: We thank the referee for pointing out that quantitative analysis of the oscillation is required in this section. We added average periods to the last figure (new figure 8, old figure 6). However, considering that the main point of this paper is combining autocatalysis and oscillation with unique chemistry of guanidines, that this has been strengthened by responses to the previous and to the next comments, and that the manuscript is already lengthy we decided to not further expand the section about the mixed oscillator.

Changes to the text:

Modified figure 8

In conclusion, the authors demonstrate a list of improvements that are promising for future engineering of such networks but it is unclear whether any of these improvement or their sum is sufficient for publication in Nature Communications. A clear-cut element of novelty would have been to actually realize any of what is proposed in the last phrase of the introduction (“Having guanidines as an output of the oscillator opens possibilities to couple the oscillator’s output to the formation of polyelectrolyte assemblies...”), but this is not done in the present work.

Our Response: The full last sentence of the introduction is “ Having guanidines as an output of the oscillator opens possibilities to couple the oscillator’s output to the formation of polyelectrolyte assemblies (e.g., polyguanidines with polyphosphates) and to the downstream formation of biologically active heterocycles (e.g., folic acid, caffeine, tetrodotoxin) and heterocycles involved in specific molecule recognition (e.g., guanine, cytosine, uracil, and their analogs) all of which can be formed by cyclization and hydrolysis of guanidines.^{45,46}. Since this paper mainly focuses on the covalent chemistry of guanidines, we decided to couple the formation of the dihydropyrimidine based heterocycle **22**, which is structurally related to uracil and is capable for H-bonding as double hydrogen bond donor, to the oscillatory process. For this experiment, we designed an oscillator based on the nitrile containing disulfide **18**. Results of these experiments are shown below.

The Title, abstract, and discussion section were modified to reflect new experimental data added to the manuscript

Changes to the text:

Figure 7 was added:

Figure 7. Kinetic studies of the thionium salt-based oscillator that uses disulfide **18**. Experimental conditions: H₂O, 1 M PB pH 8, at 40 °C; [**8**] = 56 mM; [**18**] = 56 mM; [Maleimide] = 8 mM; [Acrylamide] = 112 mM; $f/V = 1.61 \cdot 10^{-4} \text{ s}^{-1}$. **a.** Experimental data showing sustained oscillations in the concentration of thiols. **b.** Experimental data showing relative changes in the amount of **22** in CSTR. Points show normalized integrals of the HPLC peak (4.4 min, SILEC Primereser 500 4.6x250 mm, H₂O(0.1 % TFA)/CH₃CN) corresponding to **22** obtained by analysis of the drops that come from CSTR. The detection was performed by MS detector set to $m/z = +156$.

Figure 3 was modified to implement the oscillator from the disulfide 18.

Title

Text added: Autocatalytic and Oscillatory Reactions that Form Substituted Guanidines and their Cyclic Derivatives

Abstract

Text removed: Moreover, we combined thioester and thiouronium salt-based chemistries to obtain a chemical oscillator with unique responsiveness to chemical cues.

Text added: By using nitrile-containing starting materials, we constructed an oscillator where the concentration of a bicyclic derivative of dihydropyrimidine oscillates. Moreover, the mixed thioester and thiouronium salt-based oscillator showed unique responsiveness to chemical cues.

P20

Text added: **Oscillatory formation of the bicyclic pyrimidine derivative 22.** The disulfide **18**, which we used to study cascade cyclizations that follow autocatalysis, opened the possibility to couple oscillations to the formation of the bicyclic pyrimidine derivative **22**. To make an oscillator from **18**, we used the same setup as for the oscillator from **17** (Fig. 3), but we introduced several changes aimed at increasing the contribution of the intramolecular cyclization of **20** to the reaction network. First, we used equimolar amounts of **8** and **18** to increase the fraction of **20** in the dynamic mixture of thiols. Second, we decreased the amount of acrylamide to decrease the competition between cyclization of **20** and its reaction with acrylamide. Third, we increased temperature, which particularly accelerates cyclizations, to 40 °C. The increase in temperature was also necessary to increase the rate of nonautocatalytic production of thiols, which dropped in comparison with the oscillator from **17**.

The oscillator demonstrated excellent stability over 12 hours run (Fig. 7a). Therefore, we disconnected the flow cell for detection of thiols and began collecting drops directly from the outlet of CSTR for analysis by HPLC-MS. The analysis showed that amount of the bicycle **22** in CSTR changes periodically with an expected zigzag profile (Fig. 7b). It quickly rises because of the production from **20** during an active oscillation and then slowly drops because of washout during a lag phase.

Interestingly, the cyclization of **20** and the formation of **22** are part of the oscillatory network as additional negative feedback (Fig. 3). Thus, the generation of oscillations of **22** does not require a reaction in a separate compartment or an introduction of a parasitic pathway into the oscillatory network, but only stabilizes the oscillator.

P25

Text added: In this work, we have already shown how by incorporating a cascade cyclization into an oscillatory network the oscillations of a dihydropyrimidine-based heterocycle can be created. We anticipate that the autocatalytic and oscillatory reaction networks that involve close analogs of nucleic bases or that generate polyelectrolytes with guanidinium cations can now be designed and synthesized.

Minor comment:

P8: "The graph shows the expected negative correlation between the pKa of a leaving group and the rate of nucleophilic substitution": replace 'correlation' by 'trend', correlation is a statistical quantity.

Our Response: Corrected

Changes to the text:

P8

Text removed: The graph shows the expected negative correlation between the pKa of a leaving group and the rate of nucleophilic substitution;

Text added: The graph shows the expected negative trend between the pKa of a leaving group and the rate of nucleophilic substitution;

Complete, Unfragmented Referees' Comments

Reviewer #1 (Remarks to the Author):

Cyclic chemical processes arise naturally and play an important role in the transformation of chemical energy into work in soft matter systems. There are indeed limited examples of synthetic organic oscillators. Here the authors introduce a new chemical oscillator involving thiouronium salts and formation of guanidines. This is a really beautiful piece of work, carefully performed and well characterized. It is certainly novel and this oscillator may have interesting applications. I believe this manuscript should be published.

Line 174 It's not clear to me why use addition of mercaptoethanol to eliminate the induction period, why not add the autocatalyst 1?

Line 226, I would probably say residence time rather than species lifetime which is a term often associated with the kinetics.

Figure 5 has a particularly long caption, might be clearer if some of the detail was removed if it is in the main text eg how the data is obtained.

Line 310. Systems with only three variables can show all kinds of interesting behavior, including chaos. This might be a subcritical Hopf, usually difficult to stabilize experimentally. Then it is a testament to the carefully controlled experimental set-up. Perhaps better to say you were surprised to observe it in this new organic system or something similar.

Reviewer #2 (Remarks to the Author):

The manuscript of Semenov and coworkers, entitled "Autocatalytic and Oscillatory Reactions that Form Substituted Guanidines" presents very interesting new versions or more precisely a new subclass of the previously published thiol autocatalysis based organic oscillators (Semenov, S., Kraft, L., Ainla, A. et al. Autocatalytic, bistable, oscillatory networks of biologically relevant organic reactions. *Nature* 537, 656–660 (2016).). The manuscript is well written, the figures help the reader to understand the chemistry and to follow the experimental procedures. The applied methodology is appropriate, the use of CSTR is a standard tool to get chemical oscillations. The description of the experiments is detailed enough to reproduce the results.

Questions and comments:

1) Please point out the main differences of these new systems and the previously published thiol autocatalysis based organic oscillators (Semenov, S., Kraft, L., Ainla, A. et al. Autocatalytic, bistable, oscillatory networks of biologically relevant organic reactions. *Nature* 537, 656–660 (2016).) Does the mechanism of the autocatalysis change with the replacement of the previously used L-alanine ethyl thioester by thiouronium salts? I find the mechanism presented in Figure 1 in the previous paper (*Nature* 537, 656–660) quite similar to the one in this manuscript in Figure 3. Do these systems belong to the same class or are they inherently different?

2) The design presented here follows the design method, which has theoretically confirmed (e.g. [J. Boissonade and P. De Kepper, Transitions from bistability to limit cycle oscillations. Theoretical analysis and experimental evidence in an open chemical system, *J. Phys. Chem.*, 84

(1980), 501–506.] and [G. Dangelmayr and J. Guckenheimer, On a four parameter family of planar vector fields,

Archive for Rational Mechanics and Analysis, 97 (1987), 321–352.]) and has been successfully used since the '80s to design a various class of chemical oscillators (I. R. Epstein, K. Kustin, P. De Kepper and M. Orbán, "Oscillating Chemical Reactions," Sci. Amer. 248(3), 112-123 (1983).). Not only the experimental strategy but the suggested model and simulations presented in the supporting material shows directly this connection. My opinion is, that this fact must be mentioned in the introduction.

This design method is based on the fact that a bistable (autocatalytic) system can be readily changed into an oscillatory system by the addition of an appropriate feedback reaction. Now, bistability is a common phenomenon when an autocatalytic reaction is performed in a CSTR. Do the authors observe bistability between the stationary states of the CSTR content?

In the light of the above-mentioned theoretical works the observed bistability between an oscillatory state and a steady-state is not surprising, but the consequence of the general dynamics of these types of networks. For instance, this can be the sign of a subcritical Hopf-bifurcation. This phenomenon is well known and has been observed in different inorganic oscillators, see e.g. M. Orban Oscillations and bistability in the copper(II)-catalyzed reaction between hydrogen peroxide and potassium thiocyanate J. Am. Chem. Soc. 1986, 108, 22, 6893–6898.

3) In the introduction the authors state, that "Thioesters-based oscillators remain the only example of purely organic oscillators". Later on the work of th A. Taylor group, entitled "An Organic-Based pH Oscillator" is cited. Can we consider the systems of base-catalyzed dehydration of a hydrated carbonyl compound as organic oscillators or not? There are also reports on oscillations during the Maillard reaction, e.g. Voeikov, V.L., Koldunov, V.V. & Kononov, D.S. New Oscillatory Process in Aqueous Solutions of Compounds Containing Carbonyl and Amino Groups¹. Kinetics and Catalysis 42, 606–608 (2001). I think these systems are also organic reaction-based chemical oscillators. If the authors agree, than introduction should be refined.

We refined the sentence. O₂ oxidation reactions are not homogenous so we decided to not include. Modification of components is point in introduction.

4) I agree with the statement of the authors, that the inorganic chemistry based oscillators and the biochemical oscillators are quite different. But, I do not think, that the main point is the strong oxidizing nature or the extreme pH of the inorganic systems (in fact it is not always the case). More significant is the difference in the mechanism of the oscillations, which have been pointed out by Novák and Tyson (Design principles of biochemical oscillators. Nat Rev Mol Cell Biol 9, 981–991). The dominant mechanism of inorganic chemistry based oscillators is direct autocatalysis (nonlinear positive feedback). However, this is rarely the case in biochemistry. In biochemistry oscillations are typically based on delayed negative feedback loops. Considering it, I would say the mechanism of these organic oscillators are much closer (or even the same) to the one of the inorganic oscillators, since it is based on direct autocatalysis (see the supporting file S+A->2A). I would like to ask the authors to consider and analyze this mechanistic aspect also in their paper.

5) I would suggest not to call a single cycle (a pulse) to oscillations like in "S4.3. Batch experiments showing single oscillation".

Finally, I want to mention again that the experimental work is quite impressive and this work is an important development in the field of oscillatory chemical reactions.

Reviewer #3 (Remarks to the Author):

The work by Semenov et al. describes the development of autocatalytic and oscillating systems by rational design of chemical reactions. The manuscript is well written, the study is well done, and the topic is of interest to a broad readership, including life-like materials and origins of life. However, the level of novelty of this work is questionable as it is a variation of references [39] and [40] of the last author. Compared to these publications, the innovations are:

- A. replacing thioesters by thiouronium salts as feedstock of the autocatalytic reaction;
- B. "demonstrate the possibility of conversion of guanidines formed in the autocatalytic reaction into analogs of nucleobases";
- C. replacing maleimide by $K_3[Fe(CN)_6]$ as an inhibitor for oscillations, and play with the various chemicals to perturb the dynamics of the oscillators.

Concerning point A, the expansion was implemented through minor structural changes (amide-to-amidine, O-to NH substitution). Note that modification of the feedstock is done in reference [39], but with other types of thioesters only. The reaction between S-substituted amidines and amines that yield guanidines is not surprising. Such bioconjugation applications in itself pertains to a more specialized journal. This point however enables point B.

Concerning point B, guanidine is undoubtedly a more versatile precursor for further functionalization. A first problem is that the example of 2-amino-6-methylpyrimidin-4(3H) formation is not sufficient to fully appreciate the functional potential of this building block. A second problem is that this reaction is done separately from the autocatalytic reaction (chemical '18' is resynthesized) and there is no demonstration of both occurring together. In this sense, the claim of the authors that they "demonstrate the possibility..." is misleading. What is shown at this stage is rather a preliminary study toward showing autocatalysis coupled to reactions yielding products of interest.

Concerning point C, if the tunability of the system by external cues is to be the main claim of the paper, the changes in amplitude and frequency should be quantified (these changes are only visually displayed and left to the judgement of the reader) and such quantification should be done for a range of conditions that is systematic enough to appreciate the generality and limits of modulations (e.g. similarly to what the authors do in Fig. 5d for the existence of oscillations, or in Fig. 3c of their reference

[39]).

In conclusion, the authors demonstrate a list of improvements that are promising for future engineering of such networks but it is unclear whether any of these improvements or their sum is sufficient for publication in Nature Communications. A clear-cut element of novelty would have been to actually realize any of what is proposed in the last phrase of the introduction (“Having guanidines as an output of the oscillator opens possibilities to couple the oscillator’s output to the formation of polyelectrolyte assemblies...”), but this is not done in the present work.

Minor comment:

P8: “The graph shows the expected negative correlation between the pKa of a leaving group and the rate of nucleophilic substitution”: replace ‘correlation’ by ‘trend’, correlation is a statistical quantity.

Reviewer #1 (Remarks to the Author):

I think that the authors have addressed the referees comments and the revised manuscript is publishable.

Reviewer #2 (Remarks to the Author):

The authors properly addressed my critics. The revised manuscript can be accepted as is.

Reviewer #3 (Remarks to the Author):

The authors have fully addressed my concerns. In particular, there is now a clear demonstration of coupling of autocatalysis with a downstream functionalization. That this can be rationally designed and successfully implemented in a small molecule chemistry is remarkable.

Philippe Nghe